# Your Image is Secretly the Last Frame of a Pseudo Video

## Abstract

Diffusion models, which can be viewed as a special case of hierarchical variational autoencoders (HVAEs), have shown profound success in generating photo-realistic images. In contrast, standard HVAEs often produce images of inferior quality compared to diffusion models. In this paper, we hypothesize that the success of diffusion models can be partly attributed to the additional self-supervision information for their intermediate latent states provided by corrupted images, which along with the original image form a pseudo video. Based on this hypothesis, we explore the possibility of improving other types of generative models with such pseudo videos. Specifically, we first extend a given image generative model to their video generative model counterpart, and then train the video generative model on pseudo videos constructed by applying data augmentation to the original images. Furthermore, we analyze the potential issues of first-order Markov data augmentation methods, which are typically used in diffusion models, and propose to use more expressive data augmentation to construct more useful information in pseudo videos. Our empirical results on the CIFAR10 and CelebA datasets demonstrate that improved image generation quality can be achieved with additional self-supervised information from pseudo videos.

## 1 Introduction

Sequential models form a popular framework for generating images (Gulrajani et al., 2017; Sønderby et al., 2016a; Ho et al., 2020; Liu et al., 2022; Albergo et al., 2023; Lipman et al., 2023; Shi et al., 2023; Wang et al., 2024). Instead of generating images from noise in one shot, which can be challenging, these models gradually transform noise into images using multiple intermediate steps. Among them, diffusion models (Sohl-Dickstein et al., 2015; Ho et al., 2020; Song et al., 2021b) and their variants (Kingma et al., 2021; Nichol & Dhariwal, 2021; Song et al., 2021a; Rissanen et al., 2022; Bansal et al., 2023; Hoogeboom & Salimans, 2023) have shown impressive ability to generate high quality images in recent years.

While diffusion models can be viewed as a special case of a traditional sequential generative model, i.e., hierarchical variational autoencoders (HVAEs) (Sønderby et al., 2016a; Maaløe et al., 2019; Vahdat & Kautz, 2020), they tend to outperform standard HVAEs significantly in practice. The major differences between diffusion models and standard HVAEs are two-fold. First, diffusion models tend to have much more intermediate states, which may help improve the generation quality (Huang et al., 2021). Second, diffusion models incorporate exact self-supervised information for their intermediate states: they are supposed to match the corrupted (e.g., noisy or blurred) versions of the original target image at different corruption levels. These additional information helps regularize training and guide generation in diffusion models. On the other hand, the intermediate states in standard HVAEs are unobserved and one does not have explicit control of them. Consequently, there may be many different distributions over the intermediate states that are capable of generating images (i.e., the issue of unidentifiability (Locatello et al., 2019; Khemakhem et al., 2020)). The lack of identifiability of the intermediate states may pose challenges to the optimization during training since it suggests a huge hypothesis space with many sub-optimal solutions.

In this paper, we hypothesize that incorporating such self-supervised information into flexible generative models, as in diffusion models, may be one of the key reasons that they achieve good generation performance. Based on this assumption, we explore the possibility of improving other types of image generative models by extending them to video generative models and artificially injecting self-supervised information in the form

of pseudo videos whose frames are created by applying data augmentation to the original images. These pseudo videos are then used to train our video generative models. After that, we compare the generation quality of the last frame of the pseudo video (corresponding to the original image) generated by the video generative model with that of the images generated by the original image generative model. Empirically, we observe improved image generation quality via pseudo video generation compared to the images directly generated by the original image generative model. Theoretically, we provide intuitions on why designing better pseudo videos with data augmentation beyond first-order Markov chains can be helpful.

The contributions of our paper are summarized below.

- (Section 2) Our key insight is that pseudo videos created by corrupting the original target image may provide useful self-supervised information for training generative models. This is demonstrated by a comparison between diffusion models and standard HVAEs as a motivating example.

- (Sections 3 and 4) We attempt to improve two popular generative model frameworks, VQVAE (Van Den Oord et al., 2017) and Improved DDPM (Nichol & Dhariwal, 2021), by extending them to their video generative model counterparts and training them on pseudo videos. Empirically, we show that this procedure improves the image generation quality with pseudo videos of just a few frames. In general, our proposed framework provides a new way of scaling up any image generative model with its video generative model counterpart for improved performance.

- (Section 4) Theoretically, we analyze the potential issue of certain pseudo videos, including those in the form of first-order Markov chains, in autoregressive video generation frameworks. Based on our theoretical results, we propose a simple and effective approach which avoids the potential issues by constructing higher-order Markov pseudo videos.

## 2 Motivation

### 2.1 Preliminaries: Sequential Generative Models

Let $x \in \mathbb{R}^n$ be an observed variable of interest. The task of generative modeling aims to fit a parametric model $p_\theta(x)$ to estimate the data distribution $p(x)$ using samples from $p(x)$.

Hierarchical variational autoencoders (HVAEs) (Sønderby et al., 2016a; Maaløe et al., 2019; Vahdat & Kautz, 2020) employ a sequence of latent variables $x_1, \cdots, x_T \in \mathbb{R}^d$ ($d \le n$) to capture low dimensional representations of the data $x_0 := x$ at different fidelity (Salimans, 2016):

$$p_\theta(x_0) = \int p(x_T) \prod_{t=1}^{T} p_\theta(x_{t-1}|x_t) dx_{1:T}, \tag{1}$$

where the prior distribution over the last latent variable $x_T$ is often set to standard Gaussian $p(x_T) = \mathcal{N}(x_T|0, I)$, and the means of the likelihoods (or generation models) are parameterized by neural networks:

$$p_\theta(x_{t-1}|x_t) = \mathcal{N}(x_{t-1}|\mu_\theta(x_t, t), \sigma_t^2 I). \tag{2}$$

HVAEs approximate the intractable posterior

$$p_\theta(x_{1:T}|x_0) = \frac{p(x_T) \prod_{t=1}^{T} p_\theta(x_{t-1}|x_t)}{p_\theta(x_0)} \tag{3}$$

with a variational distribution (or inference model) $q_\phi(x_{1:T}|x_0)$. Different design choices for the factorization of the inference model have been proposed, including "bottom-up" factorization (Burda et al., 2015):

$$q_\phi(x_{1:T}|x_0) = \prod_{t=1}^{T} q_\phi(x_t|x_{t-1}), \tag{4}$$

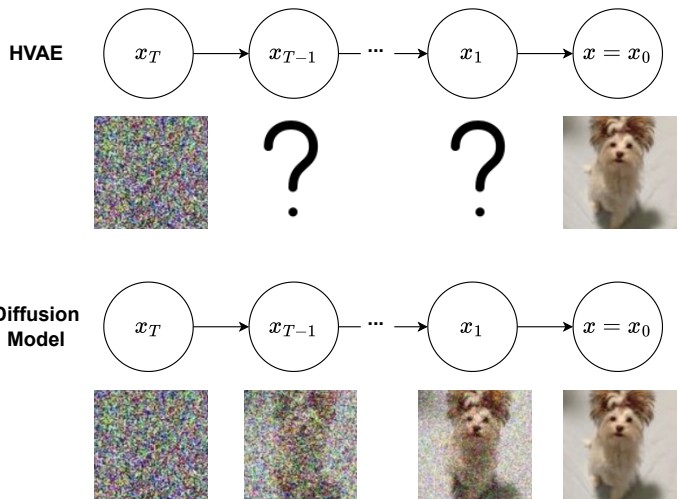

Figure 1: Diffusion model vs HVAE: compared to standard HVAEs, diffusion models incorporate inductive bias for its intermediate latent states with self-supervised signals.

and "top-down" factorization (Sønderby et al., 2016a):

$$q_\phi(x_{1:T}|x_0) = q_\phi(x_T|x_0) \prod_{t=1}^{T} q_\phi(x_{t-1}|x_t, x_0), \tag{5}$$

where the factors $q_\phi(x_t|x_{t-1})$ and $q_\phi(x_{t-1}|x_t, x_0)$ are mean-field Gaussian distributions with mean and diagnoal variance parameterized by neural networks. Notably, HVAEs only require the prior over the last latent variable be a fixed distribution (e.g., standard Gaussian) as shown in Figure 1 and let the model figure out all the intermediate latent variables by maximizing the tractable evidence lower bound (ELBO) with respect to the parameters of both generation and inference models:

$$\max_{\theta,\phi} \mathcal{F}(\theta, \phi) = \mathbb{E}_{q_\phi(x_{1:T}|x_0)} \left[ \log \frac{p(x_T) \prod_{t=1}^{T} p_\theta(x_{t-1}|x_t)}{q_\phi(x_{1:T}|x_0)} \right] \leq \log p_\theta(x_0). \tag{6}$$

Such flexibility makes HVAEs very expressive but also very difficult to train in practice (Kingma et al., 2016; Sønderby et al., 2016b), despite the efforts of restricting the flexibility of the network architectures such as sharing the parameters of the inference and generation models as in the top-down inference model (Sønderby et al., 2016a; Vahdat & Kautz, 2020; Child, 2021).

Diffusion models (Sohl-Dickstein et al., 2015; Ho et al., 2020; Song et al., 2021b), on the other hand, have achieved state-of-the-art generation performance for images. Similar to HVAEs, diffusion models also employ a sequence of latent variables in the generation process (or denoising process) as in Eq. 1 with a similar parameterization to that in Eq. 2. However, unlike HVAEs, diffusion models define a fixed "bottom up" inference model (or diffusion process):

$$q(x_{1:T}|x_0) = \prod_{t=1}^{T} q(x_t|x_{t-1}) = \prod_{t=1}^{T} \mathcal{N}(x_t|\sqrt{\alpha_t}x_{t-1}, (1-\alpha_t)I), \tag{7}$$

where $q(x_t|x_{t-1})$ are pre-defined Gaussian convolution kernels, and no dimensionality reduction is performed (i.e., $d = n$). Diffusion models are also trained by maximizing the ELBO but only with respect to the parameters $\theta$ of the generation model. Due to the simple form of the inference model, Ho et al. (2020) shows that the "top-down" form of the inference model is analytically tractable with the form:

$$q(x_{t-1}|x_t, x_0) = \mathcal{N}(x_{t-1}|\tilde{\mu}(x_t, x_0), \tilde{\beta}_t^2), \tag{8}$$

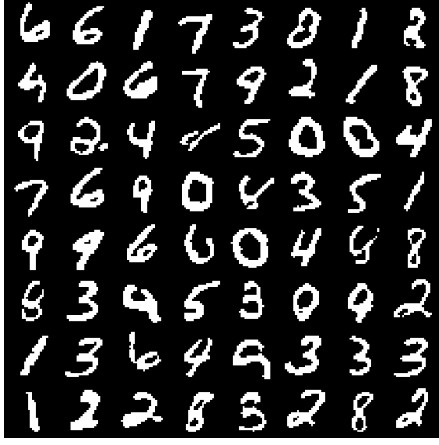

(a) HVAE with heat equation encoder.

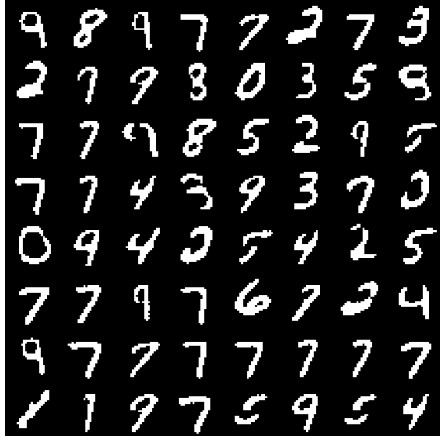

(b) Standard HVAE with learnable encoder.

Figure 2: Generated digits from HVAE with encoder fixed according to the heat equation and standard HVAE with learnable encoder. Both HVAEs use the same decoder architecture as in Rissanen et al. (2022)

where $\tilde{\mu}$ and $\tilde{\beta}_t$ have analytical solutions with closed-form expressions. As a result, the ELBO of diffusion models can be further simplified using variance reduction tricks:

$$\max_{\theta} \mathcal{F}(\theta) = \mathbb{E}_{q(x_{t-1}|x_t,x_0)}\left[\log p_{\theta}(x_0|x_1) - \sum_{t=1}^{T} \mathrm{KL}(q(x_{t-1}|x_t,x_0)||p_{\theta}(x_t|x_{t-1}))\right] \tag{9}$$

$$= \mathbb{E}_{q(x_{t-1}|x_t,x_0)}\left[\log p_{\theta}(x_0|x_1) - \sum_{t=1}^{T} \frac{\|\mu_{\theta}(x_t,t) - \tilde{\mu}(x_t,x_0)\|^2}{2\sigma_t^2}\right]. \tag{10}$$

## 2.2 Diffusion Model vs Hierarchical VAE

Compared to the objective for training standard HVAEs (Eq. 6), one can see that the objective for training diffusion models (Eq. 10) incorporates direct control for the intermediate states. Due to the fixed pre-defined inference model, the objective in Eq. 10 is simplified. In particular, its second term suggests that at each intermediate step $t$, the mean function $\mu_{\theta}(x_t,t)$ in the generation model is trained by matching a noisy version $\tilde{\mu}(x_t,x_0)$ of the original target image $x_0$. In contrast, the objective for standard HVAEs has no such information to impose any control over their intermediate states since their inference models are parameterized by flexible neural networks and keep being updated along with the generation model by end-to-end training. Figure 1 illustrates this difference between diffusion models and standard HVAEs. Without the aid of the self-supervised information, the intermediate states in standard HVAEs are very flexible, which implies that they are unidentifiable in the sense that there could be many plausible distributions over them that can generate images (i.e., many sub-optimal solutions), which makes the optimization harder as $T$ becomes larger. In contrast, diffusion models may benefit from the self-supervised information (i.e., noisy images) for their intermediate states, for which optimization can be less challenging even with large $T$, since this inductive bias pins down one specific route of generation, which eliminates other solutions that are inconsistent with this inductive bias.

Table 1: Inception Score (IS) of generated digits from HVAE with encoder fixed according to the heat equation and standard HVAE with learnable encoder.

|  | HVAE with heat equation encoder | Standard HVAE with learnable encoder |
|---|---|---|
| IS | **9.32** | 7.27 |

To show the critical role of self-supervised information, we train an HVAE with a similar architecture as in Rissanen et al. (2022) on the binarized MNIST dataset (Salimans et al., 2015) as a proof of concept, where

the encoder is fixed according to the heat equation (Rissanen et al., 2022):

$$q(x_{1:T}|x_0) = \prod_{t=1}^{T} q(x_t|x_0) = \prod_{t=1}^{T} \mathcal{N}(x_t|F_h(t)x_0, \sigma_h^2), \tag{11}$$

where $F_h$ is the matrix for simulating the heat equation until time $t$. This can be seen as an HVAE trained with explicit supervision signals from pseudo videos created by the heat equation. We create $T = 18$ frames of pseudo videos for each training image. Figure 2 demonstrates that the HVAE trained with pseudo videos created by the heat equation can generate much sharper and diverse digits than a standard HVAE which uses the same architecture but with a learnable encoder. We also report in Table 1 the Inception Score (IS) of the generated images for quantitative comparison and HVAE trained with pseudo videos created by the heat equation achieves noticeably higher IS than standard HVAE. Here, we deliberately use heat equation instead of Gaussian noise to create pseudo frames to show that there are plenty of choices to create pseudo videos that contain useful self-supervised information besides adding Gaussian noise as in standard diffusion models.

## 2.3 Improving Image Generation via Pseudo Video Generation

With the concept of pseudo videos and their effectiveness in diffusion models, we are interested in the following open generic question in this paper:

*Is it possible to improve other types of image generative models by jointly modelling the distribution of the original image and its corresponding pseudo video which contains self-supervised information?*

The answer is affirmative. In this paper, we show empirical evidence of the advantages of pseudo videos on two types of generation models, namely improving VQVAE (Van Den Oord et al., 2017) (Section 3) and DDPM (Nichol & Dhariwal, 2021) (Section 4.2) with Phenaki (Villegas et al., 2022) and Video Diffusion (Harvey et al., 2022) trained on pseudo videos, respectively. Moreover, we provide theoretical arguments favouring the use of more expressive ways of creating pseudo videos in the autoregressive video generation framework (Section 4.1), beyond the first-order Markov strategy as typically used in the forward process of standard diffusion models.

## 3 Improved Reconstruction and Generation in VQ-VAE with Pseudo Videos

In this section, we utilize pseudo videos to improve image generation quality of Vector Quantized Variational Autoencoder (VQVAE) (Van Den Oord et al., 2017), where the latent variables $z$ are discrete tokens. Specifically, we employ its video generative model counterpart C-ViViT (Villegas et al., 2022) to compress the pseudo videos into latent discrete tokens. C-ViViT is trained by reconstructing the pseudo videos with L2 reconstruction loss (Kingma & Welling, 2013), Vector Quantization (VQ) loss (Van Den Oord et al., 2017), GAN style adversarial loss (Karras et al., 2020), and image perceptual loss (Johnson et al., 2016; Zhang et al., 2018). For the latent space created by a C-ViViT, we consider two generative models to fit a prior $p(z)$ for the latent tokens:

- VideoGPT (Yan et al., 2021) uses an autoregressive (AR) Transformer (Brown et al., 2020) to factorize $p(z) = \prod_{i=1}^{d} p(z_i|z_{<i})$ in an autoregressive manner with masked self-attention, where $d$ is the total number of the tokens, and is trained with maximum likelihood. VideoGPT is the video generative model counterpart extended from ImageGPT (Chen et al., 2020a).

- Phenaki (Villegas et al., 2022) uses a bidirectional Transformer (Vaswani et al., 2017) to predict all tokens in one shot rather than in an autoregressive manner. At each training step, one samples a masking ratio $\gamma \in (0, 1)$, and the model is trained by predicting the masked tokens given the unmasked ones. During generation, all tokens are masked initially, and the model predicts all tokens simultaneously. The generation will then be refined following a few steps of re-masking and re-prediction, with a decreasing masking ratio as we proceed. Phenaki is the video generative model counterpart extended from MaskGit (Chang et al., 2022).

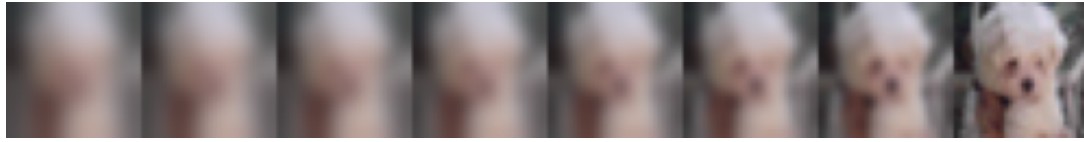

Figure 3: An example of pseudo video constructed by transforming an image of a dog using blurring.

We compare the generation quality of the last frames (corresponding to the original images) in the generated videos from the video generative model trained on pseudo videos to the images generated by the original image generative model trained on the original target images.

**Datasets.** We create 8-frame and 18-frame pseudo videos using images from two benchmark datasets, CIFAR10 ($32 \times 32$) (Krizhevsky et al., 2009) and CelebA ($64 \times 64$) (Liu et al., 2015), with the blurring technique from Bansal et al. (2023). To create 8-frame pseudo videos, we blur the images recursively 7 times with a Gaussian kernel of size $11{\times}11$ and standard deviation growing exponentially at the rate of 0.05. For 18-frame pseudo videos, we blur the images 17 times with same Gaussian kernel but with a standard deviation growing exponentially at the rate of 0.01. The pseudo videos are organized such that the last frames are the original target images and the first frames are the blurriest images. Figure 3 shows an example of such a pseudo video. We use 1-frame to denote images generated by the original image autoencoder/generative model trained on the original target images.

**Network Architectures.**[1] We train VQVAE based generative models with a codebook size of 1024 for the discrete latent tokens. Specifically, we use C-ViViT as compression model (autoencoder) for pseudo videos. For a pseudo video with shape $(T, H, W, C)$, we compress it to discrete latent tokens with shape $(\frac{T}{2}, \frac{H}{4}, \frac{W}{4}, C)$. For images with shape $(1, H, W, C)$, we use VQVAE to compress them to tokens with shape $(1, \frac{H}{4}, \frac{W}{4}, C)$. We then consider two video generative models, VideoGPT and Phenaki (and their image generative model counterparts, ImageGPT and MaskGit), to fit the prior over the latent tokens.

- **VQVAE/C-ViViT (reconstruction).** We use a similar architecture as in Villegas et al. (2022), which has a 4-layer spatial Transformer and a 4-layer temporal Transformer, with a hidden dimension of 512. For 1-frame VQVAE model, we consider a 8-layer spatial Transformer.

- **ImageGPT/VideoGPT (AR generation).** We use a similar architecture as in Yan et al. (2021), which has a 8-layer autoregressive (AR) Transformer with 4 attention heads and a hidden dimension of 144.

- **MaskGit/Phenaki (latent masked generation).** We use a similar architecture as in Villegas et al. (2022), which has a 6-layer bidirectional Transformer with a hidden dimension of 512.

**Evaluation Metric.** We compute Frechet Inception Distance (FID) (Heusel et al., 2017) with 50k samples to evaluate the quality of images, either from the last-frames of videos generated by video models trained on pseudo videos, or images generated by image models trained on the original target images.

**Results.** Table 2 shows the last-frame FID of C-ViViT for reconstruction (we also report Peak Signal-to-noise Ratio (PSNR) for reconstruction in Appendix C) and that of VideoGPT and Phenaki for AR and masked generation on CIFAR10 and CelebA, respectively. The 1-frame results correspond to the performance of their image model counterparts (i.e., VQVAE for image reconstruction, and ImageGPT and MaskGit for AR and masked image generation, respectively). We observe that pseudo videos indeed help improve the training of the C-ViViT as the reconstruction quality of the last frame is significantly improved with a few more frames. We show reconstructed CIFAR10 and CelebA images from different C-ViViT models in Figures 4 and 5, respectively. It can be seen that for both datasets, the reconstructed images trained with 1-frame

---

[1]For 1-frame models, we have tried using deeper architectures but observed no improvement in performance, which suggests that our pseudo video framework can help further improve the performance of generative models while simply increasing the model size becomes ineffective.

Table 2: Last-frame FID of images produced by C-ViViT (reconstruction), VideoGPT (AR generation) and Phenaki (latent masked generation) trained on pseudo videos constructed from CIFAR10 and CelebA images. 1-frame results are obtained from their image counterparts VQVAE (reconstruction), ImageGPT (AR generation) and MaskGit (latent masked generation) trained on original CIFAR10 and CelebA images.

|  | CIFAR10 | | | CelebA | | |
|---|---|---|---|---|---|---|
|  | 1-frame | 8-frame | 18-frame | 1-frame | 8-frame | 18-frame |
| Reconstruction | 84.25 | 13.81 | **11.26** | 24.62 | 5.72 | **2.27** |
| AR Generation | 91.65 | **54.60** | 69.23 | 32.98 | 30.19 | **28.08** |
| Latent Masked Generation | 89.78 | **35.50** | 47.65 | 27.34 | 16.87 | **16.66** |

models are over-smoothed and this issue is resolved by using pseudo videos which produce much sharper images. Quantitatively, reconstruction FID improves as more frames are used. Pseudo videos also improve the image generation performance compared to the 1-frame results. Interestingly, we see a diminishing return as we include more frames. For CelebA images, 18-frame pseudo videos help achieve the best image generation performance for both AR and latent masked generation, but 8-frame models achieve a comparable performance. For CIFAR10 images, 8-frame pseudo videos result in better image generation performance than 18-frame pseudo videos, which suggests the latent codes have a more complex prior distribution in order to reconstruct model pseudo videos with 18 frames well and therefore this prior is more difficult for VideoGPT or Phenaki to capture. In summary. while pseudo videos help improve the performance, the optimal number of frames may depend on the dataset and the augmentation strategy. This diminishing return is not a severe issue in practice since practitioners may prefer to improve the generation with just a few more pseudo frames to avoid introducing a high computational cost.

## 4 Improved Generation via Higher-order Markov Pseudo Videos

Since pseudo video contains extra information on the target image, we would like to better understand what type of additional information can be leveraged to achieve better image generation quality. In practice, since there are infinitely many data augmentation strategies to create pseudo videos, we would like to study which types of data augmentation are more favourable to shed light on the practical design of pseudo videos.

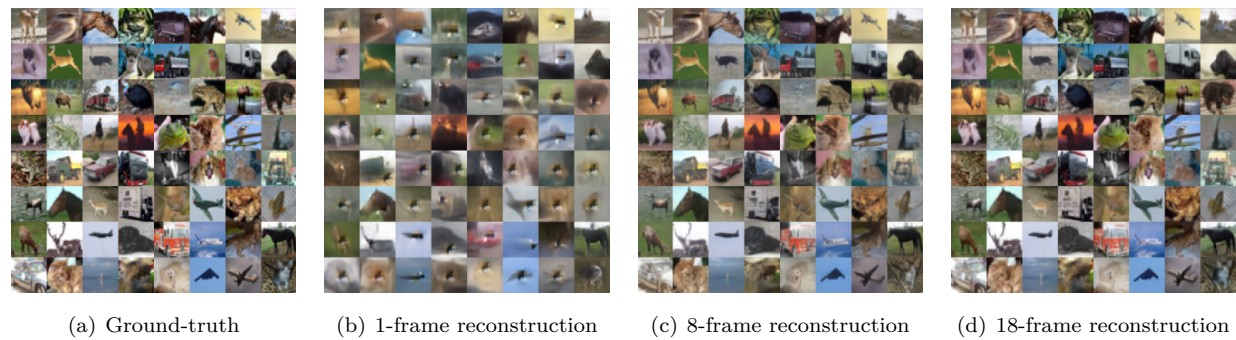

(a) Ground-truth          (b) 1-frame reconstruction          (c) 8-frame reconstruction          (d) 18-frame reconstruction

Figure 4: Ground-truth images and reconstructed images from VQVAE/CViViT trained on CIFAR10.

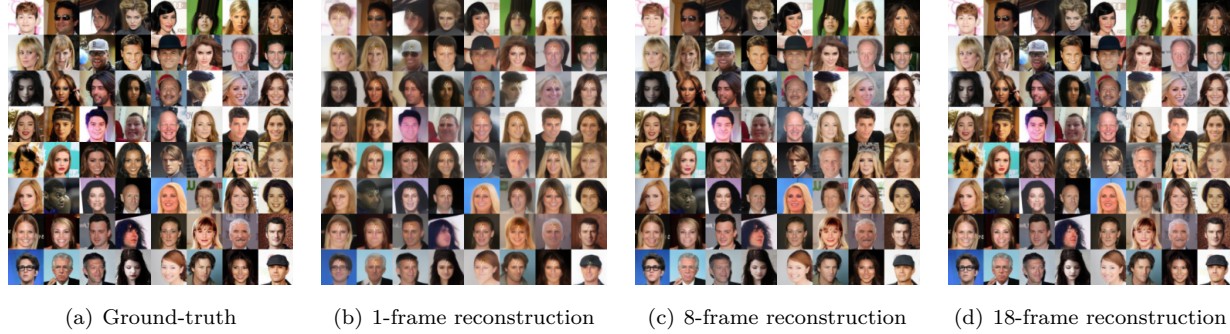

(a) Ground-truth      (b) 1-frame reconstruction      (c) 8-frame reconstruction      (d) 18-frame reconstruction

Figure 5: Ground-truth images and reconstructed images from VQVAE/CViViT trained on CelebA.

### 4.1 Is First-order Markov Chain the Optimal Choice for Creating Pseudo Videos?

Consider pseudo video $x_{1:T}$, where $x_T$ is the target image[2], and $x_t$'s $(t < T)$ are some noisy measurements of $x_T$ created with some data augmentation. We show that generative models that utilize more pseudo frames to generate $x_T$ are more likely to achieve better performance, and passing information of the target image to the pseudo frames with a first-order Markov chain as in standard diffusion models may not be the optimal choice. We demonstrate it with the following autoregressive video generation example.

Consider building a generative model $g$ that predicts $x_T$ by taking advantage of the information in $x_{T-1}$ alone. We train the model by minimizing the reconstruction error. The minimum of this loss is

$$\mathcal{L}_1^* = \min_g E_{p(x_T, x_{T-1})}[||x_T - g(x_{T-1})||_2^2] = E_{p(x_{T-1})}[\text{Var}_{p(x_T|x_{T-1})}(x_T)], \tag{12}$$

which is achieved at the non-parametric optimum $g^*(x_{T-1}) = E_{p(x_T|x_{T-1})}[x_T]$, where $\text{Var}_{p(x_T|x_{T-1})}(x_T) = E_{p(x_T|x_{T-1})}\{(x_T - E_{p(x_T|x_{T-1})}[x_T])^\top(x_T - E_{p(x_T|x_{T-1})}[x_T])\}$. Now consider another model $h$ that predicts $x_T$ using both $x_{T-1}$ and $x_{T-2}$ by minimizing the reconstruction error again. The minimum reconstruction error this time is

$$\mathcal{L}_2^* = \min_h E_{p(x_T, x_{T-1}, x_{T-2})}[||x_T - h(x_{T-1}, x_{T-2})||_2^2] = E_{p(x_{T-1}, x_{T-2})}[\text{Var}_{p(x_T|x_{T-1}, x_{T-2})}(x_T)], \tag{13}$$

which is achieved at the non-parametric optimum $h^*(x_{T-1}, x_{T-2}) = E_{p(x_T|x_{T-1}, x_{T-2})}[x_T]$. The benefit of using more pseudo frames to generate the target image can be seen from the fact that the minimum reconstruction error will never increase by using more pseudo frames since by the law of total variance,

$$\mathcal{L}_2^* - \mathcal{L}_1^* = -E_{p(x_{T-1})}\{\text{Var}_{p(x_{T-2}|x_{T-1})}(E_{p(x_T|x_{T-1}, x_{T-2})}[x_T])\} \le 0. \tag{14}$$

Moreover, the non-optimality of creating pseudo video via first-order Markov chain becomes clear: the first-order Markov data augmentation implies that $p(x_T|x_{T-1}) = p(x_T|x_{T-1}, x_{T-2})$ and consequently $\mathcal{L}_2^* = \mathcal{L}_1^*$. More specifically, for strict inequality in Eq 14, we need to avoid $p(x_T|x_{T-1}) = p(x_T|x_{T-1}, x_{T-2})$, which is equivalent to avoiding the use of either first-order Markov chain $x_T \to x_{T-1} \to x_{T-2}$ or $x_T \leftarrow x_{T-1} \to x_{T-2}$. This analysis is informative for us to design better pseudo videos, for example through data augmentation with higher-order Markov chains. We formalize the above informal reasoning into Theorem 4.1 and provide the formal proof in Appendix A.

**Theorem 4.1.** *Consider two video generative models that predict the last-frame $x_T$ some previous frames. Suppose that they take the form of $\hat{x}_T^{(g)} = g(x_{s_1}, x_{s_2}, \cdots, x_{s_k})$ and $\hat{x}_T^{(h)} = h(x_{s_1}, x_{s_2}, \cdots, x_{s_l})$, respectively, where $T > s_1 > \cdots > s_k > \cdots > s_l$. Then, we have*

$$\min_{\hat{x}_T^{(h)}} E_{p(x_T, x_{s_1}, \cdots, x_{s_l})}[||x_T - \hat{x}_T^{(h)}||_2^2] \le \min_{\hat{x}_T^{(g)}} E_{p(x_T, x_{s_1}, \cdots, x_{s_k})}[||x_T - \hat{x}_T^{(g)}||_2^2], \tag{15}$$

---

[2]Unlike diffusion models where $x_0$ denotes the original image, from here onwards we will denote the original image by $x_T$ since it is the last frame of the pseudo video.

where the equality attains if $x_T|x_{s_1}, \cdots, x_{s_k} \stackrel{d}{=} x_T|x_{s_1}, \cdots, x_{s_l}$.

**Remark 4.2.** The minimum reconstruction errors above are obtained with non-parametric optima. In practice, this corresponds to the assumption that our neural networks $g_\theta$ and $h_\phi$ are flexible enough to accurately approximate the non-parametric optima for the theorem to hold. Besides, the analysis is based on the assumption that $\{x_{s_i}\}_{i=1}^l$ are drawn from the ground-truth distribution, while in practice they also need to be generated with their associated previous frames, which means when the generated $\{\hat{x}_{s_i}\}_{i=1}^l$ are far away from their ground-truth distribution, the theorem would not hold. Nevertheless, the analysis provides intuitions of the benefit of conditional generation with longer past contexts and the potential improvement in performance by using more expressive pseudo videos rather than the ones created with first-order Markov transition as in standard diffusion models, which we empirically verify with experiments in the following section.

### 4.2 Experiments

We consider generating each frame of the pseudo videos using the information provided in the previously generated frames. In particular, we use a video diffusion model (Harvey et al., 2022) trained by predicting frames autoregressively conditioning on the most recent previous frames in a context window. We compare the performance of video diffusion models trained on pseudo videos created by both standard first-order Markov transformation and higher-order Markov transformation to empirically verify our argument in Section 4.1. We describe the detailed experimental setup below.

**Datasets.** We create 4-frame and 8-frame pseudo videos using images from CIFAR10 ($32 \times 32$) and CelebA ($64 \times 64$). We use Gaussian noise as data augmentation and we consider two strategies:

- **First-order Markov.** We add Gaussian noise recursively 3 or 7 times to create first-order Markov pseudo videos with a linear schedule (Ho et al., 2020) with $\beta$ ranging from 0.0001 to 0.05: $x_{T-t} = \sqrt{1-\beta_t}x_{T-t+1} + \sqrt{\beta_t}\epsilon,\ \epsilon \sim \mathcal{N}(0, I)$.

- **High-order Markov.** While using the same noise schedule to create $x_{T-t}$, instead of adding Gaussian noise to $x_{T-t+1}$, we use a simple strategy to create high-order Markov pseudo videos by adding Gaussian noise to the mean of $\{x_{T-t+s}\}_{s=1}^t$: $x_{T-t} = \sqrt{1-\beta_t}[\frac{1}{t}\sum_{s=1}^t x_{T-t+s}] + \sqrt{\beta_t}\epsilon$, $\epsilon \sim \mathcal{N}(0, I)$.

We plot examples of pseudo videos created with the above two strategies in Figure 6. We again use 1-frame to denote the results of the image generative model counterparts, improved DDPM (Nichol & Dhariwal, 2021), trained on the original target images. We also consider blurring as the data augmentation, however, its performance is worse than the performance of using Gaussian noise (see the **Results** paragraph below).

**Network Architectures.** We use a similar UNet architecture as in Harvey et al. (2022), with 2 residual blocks in each downsampling and upsampling layer and a base channel size of 128 across all models. Notice that Harvey et al. (2022) is built based on the same architecture as the 1-frame image diffusion model (Nichol & Dhariwal, 2021), and these hyperparameters are kept the same for the 1-frame image diffusion model. During generation, we use the "Autoreg" sampling scheme from Harvey et al. (2022) so that each frame $x_t$ is generated by conditioning on the most recently generated frames in a context window, $\{x_{t-c}\}_{c=1}^C$. The sizes of the context window $C$ (i.e., the time lag) are 2 and 4 for 4-frame and 8-frame models, respectively. We consider 1,000 diffusion steps every time we generate a new frame. Since 4-frame and 8-frame models jointly generate the first 2 and the first 4 frames (the initial context window) at the beginning, respectively, they use overall 3,000 and 5,000 diffusion steps to generate the whole pseudo videos, respectively. We also consider increasing the number of diffusion steps from 1,000 to 4,000 when training the 1-frame image diffusion model and compare it with the 4-frame video diffusion models with 3,000 diffusion steps in total to ensure the performance gain in video diffusion models is not simply because we have more diffusion steps overall.

**Results.** We again compute FID (based on 10k samples) to evaluate the models. Table 3 shows the last-frame FID of pseudo videos generated by video diffusion models for CIFAR10 and CelebA images, respectively. The 1-frame results correspond to the performance of their image counterparts (i.e., improved

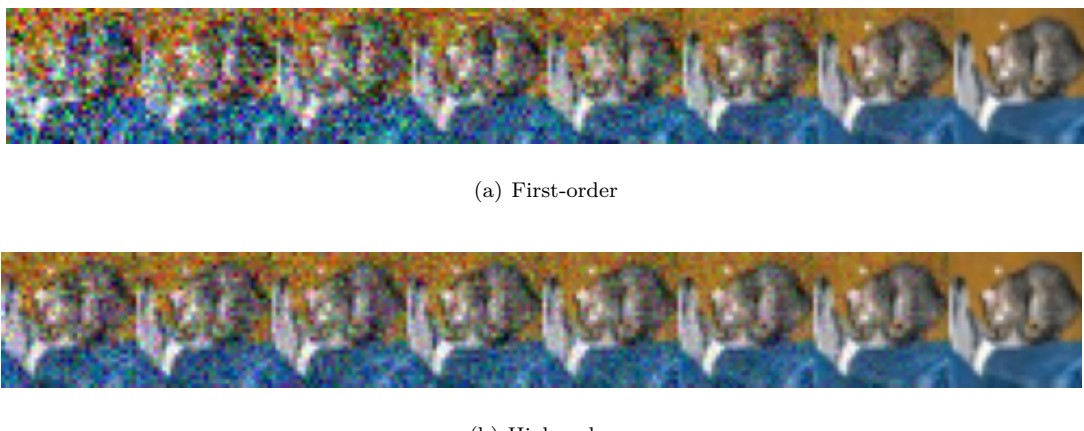

(a) First-order

(b) High-order

Figure 6: Examples of a pseudo video constructed by adding Gaussian noise to a CIFAR10 image using first-order Markov chain (top) and high-order Markov chain (bottom).

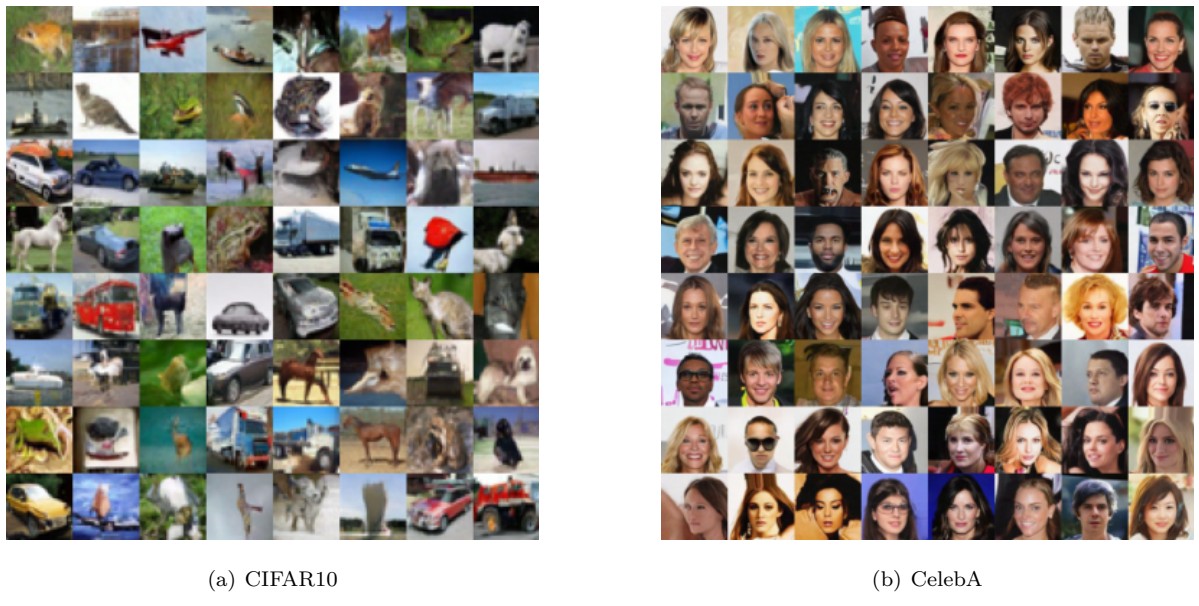

(a) CIFAR10

(b) CelebA

Figure 7: Generated images from the video diffusion models trained on 4-frame high-order Markov pseudo videos of CIFAR10 and CelebA, respectively.

DDPM). While video diffusion models trained on first-order Markov pseudo videos do not outperform the 1-frame image diffusion model, both 4-frame and 8-frame video diffusion models trained on high-order Markov pseudo videos can achieve better results on both datasets, which empirically justify the non-optimality of first-order Markov chains in terms of passing information from the target images to the pseudo frames as shown in Section 4.1, and our proposal of using more expressive pseudo videos rather than the ones created with first-order Markov chains. Notice that the 4-frame models outperform the 8-frame models, which may be due to the complex nature of the ground-truth distribution of longer pseudo videos, and thus more expressive architecture may be required to achieve optimal results (see Remark 4.2), while here we use the same UNet architecture across all models for fair practical comparison. Again, this U-turn should not be a severe issue in practice since practitioners may prefer to improve the generation with as few pseudo frames

Table 3: Last-frame FID of images generated by video diffusion models trained on pseudo videos constructed from CIFAR10 and CelebA images (with both first-order Markov or high-order Markov Gaussian noise data augmentation). 1-frame results are obtained from an image diffusion model trained on the original CIFAR10 and CelebA images with equivalent UNet architecture.

|  | CIFAR10 | | | CelebA | | |
|---|---|---|---|---|---|---|
|  | 1-frame | 4-frame | 8-frame | 1-frame | 4-frame | 8-frame |
| First-order Markov | 12.90 | 17.30 | 15.90 | 7.76 | 13.61 | 12.64 |
| High-order Markov | 12.90 | **12.58** | 12.80 | 7.76 | **6.88** | 7.55 |

as possible to reduce additional computational cost. We visualize some generated images from the 4-frame models trained on CIFAR10 and CelebA in Figure 7.

Table 4 compares the 4-frame model with an 1-frame model but with 4,000 diffusion steps. While the performance of the 1-frame model with more overall diffusion steps improves for CIFAR10 and outperforms the 4-frame model, its performance on CelebA is worse than the 4-frame model. Moreover, on CelebA, it even becomes worse than the baseline 1-frame model with only 1,000 diffusion steps while the 4-frame video diffusion model consistently improves the performance on both datasets, which suggests simply increasing the number of diffusion steps in an image diffusion model may not always be effective.

Instead of Gaussian noise, we also tried using Gaussian blur to create pseudo videos as in Section 3. However, our experiments on CIFAT10 with Gaussian blur suggest worse results than adding Gaussian noise (see Table 5), and we decided not to consider it for further experiments. This suggests that in practice the well-performed data augmentation strategies may vary across different classes of video generative models.

Table 4: Last-frame FID of images generated by video diffusion models trained on pseudo videos constructed from CIFAR10 and CelebA images with high order Markov Gaussian noise data augmentation. 1-frame results are obtained from an image diffusion model trained on the original CIFAR10 and CelebA images. Here, the 1-frame models use 4,000 diffusion steps, while the 4-frame models use 3,000 diffusion steps overall.

|  | 1-frame (1k steps) | 1-frame (4k steps) | 4-frame (3k steps overall) |
|---|---|---|---|
| CIFAR10 | 12.90 | **11.95** | 12.58 |
| CelebA | 7.76 | 7.87 | **6.88** |

Table 5: Last-frame FID of images generated by video diffusion models trained on pseudo videos constructed from CIFAR10 images with high-order Markov data augmentation (either Gaussian noise or Gaussian blur).

|  | 4-frame | 8-frame |
|---|---|---|
| Gaussian noise | **12.58** | 12.80 |
| Gaussian blur | 15.33 | 22.63 |

## 5 Related Work

### 5.1 Sequential generative Models

Hierarchical variational autoencoders (HVAEs) (Sønderby et al., 2016a; Maaløe et al., 2019; Vahdat & Kautz, 2020; Child, 2021; Xiao & Bamler, 2023) are a class of sequential generative models constructed by stacking standard VAEs (Kingma & Welling, 2013). Although HVAEs represent a rich class of expressive generative models, they are hard to train in practice due to optimization difficulty, as discussed in Section 2. Diffusion

models (Sohl-Dickstein et al., 2015; Ho et al., 2020; Song et al., 2021b; Kingma et al., 2021; Nichol & Dhariwal, 2021; Song et al., 2021a; Rissanen et al., 2022; Bansal et al., 2023; Hoogeboom & Salimans, 2023) can be seen as a special case of HVAEs where the encoders are fixed, pre-defined Gaussian convolution kernels. Specifically, they essentially regress a sequence of noisy images created from the target image with self-supervision, as described in Section 2. Despite its similarity to HVAEs, diffusion models, and latent diffusion models (Rombach et al., 2022) which apply diffusion models in the lower dimensional latent space of another latent variable model (e.g., VQVAE (Van Den Oord et al., 2017)), have achieved state-of-the-art performance partially due to the additional self-supervision signal provided by the noise-corrupted images. Flow matching (Lipman et al., 2023; Liu et al., 2022; Albergo et al., 2023; Gat et al., 2024; Wang et al., 2024) is another state-of-the-art sequential generative modelling technique that trains continuous normalizing flows (Chen et al., 2018) by regressing a sequence of vector fields inducing a probability path that connects the data distribution and prior distribution with direct self-supervision. It has been show that flow matching can learn more straight trajectories than diffusion models, which requires less number of discretization steps at generation time. Furthermore, flow matching allows us to relax the Gaussian assumption for the prior distribution and thus enables coupling between arbitrary distributions (Albergo et al., 2023). In contrast, our proposed framework introduces a new family of approaches that leverage video generative models and pseudo videos with self-supervised frames to improve any given image generative models.

### 5.2 Self-Supervised Learning

Self-supervised learning (Liu et al., 2021; Shwartz Ziv & LeCun, 2024) turns an unsupervised learning problem into a supervised learning problem by handcrafting pseudo labels for unlabeled data. There are two common approaches to self-supervised learning. 1) Contrastive learning (Chen et al., 2020b; Tian et al., 2020; Wu et al., 2020), predicts whether two inputs are different augmentations of the same original data. 2) Masked learning (Devlin et al., 2019; He et al., 2022; Fang et al., 2023) predicts randomly masked parts of an input given the unmasked parts. While our approach of fitting a video model to pseudo video sequences created by augmenting the original images does not belong to either of these families, it is essentially a new form of self-supervised learning since the pseudo video sequences can be seen as handcrafted pseudo labels for our model to predict, which provides the model with extra information (e.g., different fidelity of the original image).

## 6 Conclusion and Discussions

**Summary.** We drew our key insight from comparing standard HVAEs and diffusion models: the additional self-supervised information on the intermediate states provided by the noise corrupted pseudo frames in diffusion models may contribute to their success. Based on this insight, we proposed to leverage the self-supervised information from the pseudo videos constructed by applying data augmentation to the target images to improve the performance of image generative models. This was done by extending image generative models to their video generative models counterparts and training video generative models on pseudo videos. We show in our experiments that for two popular image generative models, VQVAE and Improved DDPM, their video generative model counterparts trained on pseudo videos of just a few frames can improve image generation performance, which empirically verified the benefit of the additional self-supervised information in the pseudo videos.

**Discussions and Future Work.** Our proposed framework provides an alternative approach of scaling up any given generative models: instead of making generative models larger by stacking more layers, we demonstrated that it was possible to improve the generation quality by turning an image generative model trained on images into its video generative model counterpart trained on pseudo videos, which is usually straightforward since many video generative models are built upon image generative models. On the other hand, this raises challenges on how to design informative pseudo videos. In autoregressive video generation frameworks, we show the potential issue of first-order Markov pseudo videos theoretically and propose to use higher-order Markov pseudo videos instead to address this issue. However, it is in general unclear what the optimal pseudo videos are within such a large design space, which we leave as a future research question. Another interesting future direction is to explore whether the same principle can be applied to

other data modalities. While self-supervised signals can be easily obtained using data augmentation for images, it remains unclear whether there are proper ways to inject self-supervised information for other data modalities, such as text or molecules.

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

## A  Proof of Theorem 4.1

*Proof.* To derive $\hat{x}_T^{(g)*}(x_{s_1}, \cdots, x_{s_k}) = \arg\min_{\hat{x}_T^{(g)}} E_{p(x_T, x_{s_1}, \cdots, x_{s_k})}[||x_T - \hat{x}_T^{(g)}||_2^2]$, we first compute the gradient,

$$
\begin{aligned}
\nabla_{\hat{x}_T^{(g)}} E_{p(x_T, x_{s_1}, \cdots, x_{s_k})}[||x_T - \hat{x}_T^{(g)}||_2^2] &= E_{p(x_{s_1}, \cdots, x_{s_k})}\{\nabla_{\hat{x}_T^{(g)}} E_{p(x_T|x_{s_1}, \cdots, x_{s_k})}[||x_T - \hat{x}_T^{(g)}||_2^2]\} \\
&= 2E_{p(x_{s_1}, \cdots, x_{s_k})}\{E_{p(x_T|x_{s_1}, \cdots, x_{s_k})}[x_T - \hat{x}_T^{(g)}]\}.
\end{aligned}
\tag{16}
$$

Setting the above gradient to 0 gives us

$$
E_{p(x_T|x_{s_1}, \cdots, x_{s_k})}[x_T - \hat{x}_T^{(g)}] = 0 \implies \hat{x}_T^{(g)*}(x_{s_1}, \cdots, x_{s_k}) = E_{p(x_T|x_{s_1}, \cdots, x_{s_k})}[x_T].
\tag{17}
$$

The minimum reconstruction error is obtained by plugging $\hat{x}_T^{(g)*}(x_{s_1}, \cdots, x_{s_k})$ in $E_{p(x_T, x_{s_1}, \cdots, x_{s_k})}[||x_T - \hat{x}_T^{(g)}||_2^2]$,

$$
\begin{aligned}
\min_{\hat{x}_T^{(g)}} E_{p(x_T, x_{s_1}, \cdots, x_{s_k})}[||x_T - \hat{x}_T^{(g)}||_2^2] &= E_{p(x_T, x_{s_1}, \cdots, x_{s_k})}[||x_T - E_{p(x_T|x_{s_1}, \cdots, x_{s_k})}[x_T]||_2^2] \\
&= E_{p(x_{s_1}, \cdots, x_{s_k})}[\text{Var}_{p(x_T|x_{s_1}, \cdots, x_{s_k})}(x_T)].
\end{aligned}
\tag{18}
$$

Similarly,

$$
\begin{aligned}
\min_{\hat{x}_T^{(h)}} E_{p(x_T, x_{s_1}, \cdots, x_{s_l})}[||x_T - \hat{x}_T^{(h)}||_2^2] &= E_{p(x_{s_1}, \cdots, x_{s_l})}[\text{Var}_{p(x_T|x_{s_1}, \cdots, x_{s_l})}(x_T)], \\
\hat{x}_T^{(h)*}(x_{s_1}, \cdots, x_{s_l}) &= E_{p(x_T|x_{s_1}, \cdots, x_{s_l})}[x_T],
\end{aligned}
\tag{19}
$$

We now show that the reconstruction error can never increase with more previous frames as inputs by observing that with $T > s_1 > \cdots > s_k > \cdots > s_l$,

$$
\begin{aligned}
\min_{\hat{x}_T^{(h)}} E_{p(x_T, x_{s_1}, \cdots, x_{s_l})}[||x_T - \hat{x}_T^{(h)}||_2^2] &= E_{p(x_{s_1}, \cdots, x_{s_l})}[\text{Var}_{p(x_T|x_{s_1}, \cdots, x_{s_l})}(x_T)] \\
&\leq E_{p(x_{s_1}, \cdots, x_{s_k})}[\text{Var}_{p(x_T|x_{s_1}, \cdots, x_{s_k})}(x_T)] = \min_{\hat{x}_T^{(g)}} E_{p(x_T, x_{s_1}, \cdots, x_{s_k})}[||x_T - \hat{x}_T^{(g)}||_2^2].
\end{aligned}
\tag{20}
$$

Indeed,

$$
\begin{aligned}
&E_{p(x_{s_1}, \cdots, x_{s_l})}[\text{Var}_{p(x_T|x_{s_1}, \cdots, x_{s_l})}(x_T)] - E_{p(x_{s_1}, \cdots, x_{s_k})}[\text{Var}_{p(x_T|x_{s_1}, \cdots, x_{s_k})}(x_T)] \\
&= E_{p(x_{s_1}, \cdots, x_{s_k})}\{E_{p(x_{s_{k+1}}, \cdots, x_{s_l}|x_{s_1}, \cdots, x_{s_k})}[\text{Var}_{p(x_T|x_{s_1}, \cdots, x_{s_l})}(x_T)] - \text{Var}_{p(x_T|x_{s_1}, \cdots, x_{s_k})}(x_T)\} \\
&= -E_{p(x_{s_1}, \cdots, x_{s_k})}\{\text{Var}_{p(x_{s_{k+1}} \cdots, x_{s_l}|x_{s_1}, \cdots, x_{s_k})}(E_{p(x_T|x_{s_1}, \cdots, x_{s_l})}[x_T])\} \qquad \text{(Law of total variance)} \\
&\leq 0.
\end{aligned}
\tag{21}
$$

Notice that if $x_T|x_{s_1}, \cdots, x_{s_k} \overset{d}{=} x_T|x_{s_1}, \cdots, x_{s_l}$, then the above difference will become 0:

$$
\begin{aligned}
&\text{Var}_{p(x_{s_{k+1}} \cdots, x_{s_l}|x_{s_1}, \cdots, x_{s_k})}(E_{p(x_T|x_{s_1}, \cdots, x_{s_l})}[x_T]) \\
&= \text{Var}_{p(x_{s_{k+1}} \cdots, x_{s_l}|x_{s_1}, \cdots, x_{s_k})}(E_{p(x_T|x_{s_1}, \cdots, x_{s_k})}[x_T]) \\
&= 0,
\end{aligned}
\tag{22}
$$

since $E_{p(x_T|x_{s_1}, \cdots, x_{s_k})}[x_T]$ is a function of $x_{s_1}, \cdots, x_{s_k}$ only.

Therefore, for strict inequality, it is necessary to avoid $x_T \perp\!\!\!\perp x_{s_{k+1}}, \cdots, x_{s_l} \mid x_{s_1}, \cdots, x_{s_k}$, which includes first-order Markov chain $(x_T \to \cdots x_{s_k} \to x_{s_l})$ as a special case.

$\square$

## B Reproducibility Statement

To our knowledge, the official implementation of Phenaki (Villegas et al., 2022) has not been released upon submission of this paper. As a result, the experiments in Section 3 are based on the implementation from `https://github.com/lucidrains/phenaki-pytorch`. The experiments in Section 4.2 are based on the official implementation of Flexible Video Diffusion Model (FDM) (Harvey et al., 2022) and Improved DDPM (Nichol & Dhariwal, 2021), from `https://github.com/plai-group/flexible-video-diffusion-modeling` and `https://github.com/openai/improved-diffusion`.

## C PSNR for Reconstruction

Table 6: PSNR of reconstruction for the last frame of pseudo videos produced by C-ViViT trained on pseudo videos constructed from CIFAR10 and CelebA images. 1-frame results are obtained from their image counterparts VQ-VAE trained on original CIFAR10 and CelebA images.

|                | CIFAR10 | | | CelebA | | |
|----------------|---------|---------|----------|---------|---------|----------|
|                | 1-frame | 8-frame | 18-frame | 1-frame | 8-frame | 18-frame |
| Reconstruction | 17.20   | 21.86   | **25.89** | 20.52   | 24.66   | **29.68** |

## D Hyperparameters

Table 7: Hyperparamters used for C-ViViT architecture and optimizer.

|                            | 1-frame      | 8-frame      | 18-frame     |
|----------------------------|--------------|--------------|--------------|
| Number of spatial layers   | 8            | 4            | 4            |
| Number of temporal layers  | -            | 4            | 4            |
| Embedding dimension        | 512          | 512          | 512          |
| Hidden dimension           | 512          | 512          | 512          |
| Number of heads            | 8            | 8            | 8            |
| Learning rate              | 1e-4         | 1e-4         | 1e-4         |
| Learning rate scheduler    | Cosine decay | Cosine decay | Cosine decay |
| Number of training steps    | 100k         | 100k         | 100k         |
| Batch size                 | 64           | 64           | 64           |

Table 8: Hyperparamters used for VideoGPT architecture and optimizer.

|                            | 1-frame      | 8-frame      | 18-frame     |
|----------------------------|--------------|--------------|--------------|
| Number of layers           | 8            | 8            | 8            |
| Embedding dimension        | 144          | 144          | 144          |
| Hidden dimension           | 144          | 144          | 144          |
| Number of heads            | 4            | 4            | 4            |
| Learning rate              | 1e-4         | 1e-4         | 1e-4         |
| Learning rate scheduler    | Cosine decay | Cosine decay | Cosine decay |
| Number of training steps    | 100k         | 100k         | 100k         |
| Batch size                 | 64           | 64           | 64           |

Table 9: Hyperparamters used for Phenaki architecture and optimizer.

|  | 1-frame | 8-frame | 18-frame |
|---|---|---|---|
| Number of layers | 6 | 6 | 6 |
| Embedding dimension | 512 | 512 | 512 |
| Hidden dimension | 512 | 512 | 512 |
| Number of heads | 8 | 8 | 8 |
| Learning rate | 1e-4 | 1e-4 | 1e-4 |
| Learning rate scheduler | Cosine decay | Cosine decay | Cosine decay |
| Number of training steps | 200k | 200k | 200k |
| Batch size | 64 | 64 | 64 |

Table 10: Hyperparamters used for UNet architecture and optimizer for Video diffusion.

|  | 1-frame | 4-frame | 8-frame |
|---|---|---|---|
| Number of downsampling/upsampling layers | 4 | 4 | 4 |
| Number of residual blocks | 2 | 2 | 2 |
| Base channel size | 128 | 128 | 128 |
| Number of diffusion steps per generating a frame | 1000 | 1000 | 1000 |
| Learning rate | 1e-4 | 1e-4 | 1e-4 |
| Number of training steps | 500k | 500k | 500k |
| Batch size | 32 | 32 | 32 |

