# OpenReview forum: "Your Image is Secretly the Last Frame of a Pseudo Video"
_TMLR — Rejected by TMLR_

### Review · Reviewer_QFcD · 2024-11-02

**Summary Of Contributions:**

The paper proposes a simple idea to improve image generation from the perspective of video generation – the process of generating an image can be thought of as a sequence of frames, or a video, where the first frame is usually not too informative (e.g., random noise) and the last frame is desired outcome (e.g., a clean datapoint from the true data distribution). The idea is to synthetically create these sequences of frames, or pseudo-videos, and learn a video generation model using these videos. The proposed approach is validated with two types of video generation models – autoregressive and masked modeling of tokens, resulting in an improved image generation compared to the 1-frame settings. Furthermore, a theoretical analysis of the benefit of using higher-order Markov chains to create the augmentations, or the pseudo-videos, is provided, with complimentary experiments.

**Audience:**

Yes

**Claims And Evidence:**

Yes

**Requested Changes:**

* The toy experiment in Section 2.2 is important and serves as motivation for why the structure, or the self-supervision, is crucial. However, some information is missing, which I suggest adding to the appendix: how is the $F_h$ matrix defined? Maybe even add a plot of several representative frames. In addition, the number $T=18$ seems random for the choice of video length, is there any justification for that or is that just what worked best? Finally, what is the dimension of the latent space $z$ in this case?
* One of the interesting points in this paper, at least to my understanding, is that in the tokenization pre-training, the synthetical augmented frames are also tokenized. I wonder what is the significance of that given that there could be infinite possibilities for corruptions, such as the Gaussian noise. What would happen if you only learned to tokenize the clean data (images) and then just tokenized the augmented frames when training the generative model? (it is non-critical, but I believe a discussion of that would strengthen this work).
* See my suggestions under “Weaknesses” above.
* See “Minor” above.
* Are the authors planning to release an open-source of the proposed approach? I believe that even though the work builds on existing video generation code bases, there is benefit in “how to create pseudo-videos”.

**Strengths And Weaknesses:**

**Strengths**:
* Informative background on approaches to design HVAEs and connecting them to diffusion models.
* Sound intuition and convincing arguments, backed with experiments.
* Theoretical and empirical analysis for using higher-order Markov chains to create the augmented versions of images.
* Simple (!)

**Weaknesses**:
* Discussion/ablation of the number of frames, why only 8 and 18?
* In general, I think this paper could benefit from additional information and discussion of hyper-parameters and computing resources (e.g., training time).
* Completeness: I find the current overview/background of utilized methods such as VideoGPT and MaskedGIT a bit lacking. I would suggest the authors add a more detailed background in the appendix for the completeness of the paper.
* Results: why measure FID for *reconstruction*? The common evaluation metrics for reconstruction are PSNR (directly related to the pixel-wise MSE), SSIM and LPIPS.
* Results: the results for the 1-frame reconstruction using VQ-VAE seem a bit weird to me. There are weird artifacts in both the CIFAR-10 and CelebA reconstruction results that I believe are the result of improper training. I found several repositories on GitHub that show much better reconstructions (at least visually), see below. Maybe something went wrong when balancing the multiple losses, perhaps the adversarial one.
* The results in Table 2 and 3 are not entirely convincing.
* The related work section on self-supervised learning seems a bit irrelevant, as the authors themselves say: “While our approach of fitting a video model to pseudo video sequences created by augmenting the original images does not belong to either of these families”. I think the more relevant works are in the masked video generation field (e.g., [1], [2]) and maybe works that focus on how to create “noisy” versions of images for diffusion modeling (e.g., [3], [4]).
* Missing some discussion of limitations: what are some limitations of this approach? E.g., training time, required computation resources, complexity and challenges of training a video generation model vs. an image generation model, and etc…

[1] Yu, Lijun, et al. "Magvit: Masked generative video transformer." Proceedings of the IEEE/CVF Conference on Computer Vision and Pattern Recognition. 2023. (https://arxiv.org/abs/2212.05199)

[2] Voleti, Vikram, Alexia Jolicoeur-Martineau, and Chris Pal. "Mcvd-masked conditional video diffusion for prediction, generation, and interpolation." Advances in neural information processing systems 35 (2022): 23371-23385. (https://arxiv.org/abs/2205.09853)

[3] Bansal, Arpit, et al. "Cold diffusion: Inverting arbitrary image transforms without noise." Advances in Neural Information Processing Systems 36 (2024). (https://arxiv.org/abs/2208.09392)

[4] Heitz, Eric, Laurent Belcour, and Thomas Chambon. "Iterative α-(de) blending: A minimalist deterministic diffusion model." ACM SIGGRAPH 2023 Conference Proceedings. 2023. (https://arxiv.org/abs/2305.03486)

GitHub repositories with implementations of VQ-VAE: https://github.com/mattiasxu/VQVAE-2 , https://github.com/vvvm23/vqvae-2 , https://github.com/rosinality/vq-vae-2-pytorch

**Minor**:
* Theorem 4.1: please clarify what are $s_1, s_2, …,s_k$ and what does it mean that $ T> s_k > s_l$.
* Equation 15: “where the equality attains if…” - I think the mathematical expression that comes after this is missing some components, or otherwise, explain what $x_T |x_{s_1},..|$ means.
* Page 9, when describing “Higher-order Markov”: I found it hard to understand how this works, I would suggest improving the wording. Perhaps provide a proper example, e.g. “for example, at frame number 5 we add the following noise…”

---

> ### Author Response · Authors · 2024-11-29
> **Response (part 1/2)**
>
> Thank you for your encouraging review and valuable feedback. We will address your questions below.
>
> >1. Discussion/ablation of the number of frames, why only 8 and 18?
>
> During our experiments, we first choose the largest number of frames to be the largest one that we can train on our GPU (they are 18 and 8 for CViViT and Video Diffusion, respectively). Then we consider including the results with the number of frames approximately being half of the maximal one (8 and 4 for CViViT and Video Diffusion, respectively).
>
> >2. More discussion of hyperparameters, computing resources, and background of utilized models.
>
> We have summarized the hyperparameter details in Appendix D in the uploaded revision. The method requires more computing resources since we need to fit a video generation model instead of an image generation model, and we will add a detailed discussion of this limitation and the background of utilized models in the final revision.
>
> >3. Evaluation metric for reconstruction.
>
> We measure reconstructed FID following [1]. Here we report PSNR as another metric suggested by the reviewer (we have included it in the uploaded revision):
>
> **Reconstruction PSNR of CVIVIT models**
> ||1-frame|8-frame|18-frame|
> |---|---|---|---|
> |CIFAR10|17.20|21.86|**25.89**|
> |CelebA|20.52|24.66|**29.68**|
>
> >4. Artefacts in the results for the 1-frame reconstruction using VQ-VAE.
>
> We follow the code in https://github.com/lucidrains/phenaki-pytorch for the training and keep the weight for adversarial loss (we use 0.1 following [2]) along with the other hyperparameters to be exactly the same across 1, 8, 18-frame models. The only difference between 1-frame model and multi-frame models is that while multi-frame models use a 4-layer spatial Transformer and a 4-layer temporal Transformer, 1-frame model uses an 8-layer spatial Transformer instead, since no temporal modelling across different frames is needed here. Notably, CNNs are usually used to build the architecture for VQVAE (including the GitHub repos sent by the reviewer) on small datasets such as CIFAR10, but here in CViViT, we are using Transformer architectures to build VQVAE because we need to take care of the temporal direction in the video. Transformers have been shown to underperform compared with CNN for small datasets (see e.g. [3,4,5]), and VQVAEs based on Transformers have only been evaluated for large datasets such as ImageNet in the literature [6,7]. One possible explanation of why multi-frame models outperform 1-frame models is that the data augmentation in the pseudo video framework implicitly increases the size of the training set, which is in favour of Transformer architectures.
>
> >5. The results in Table 2 and 3 are not entirely convincing.
>
> If the model class is fully expressive such that we can reach the theoretical optima in Theorem 4.1 for 1-frame, 4-frame and 8-frame, then the rank of the theoretically optimal performance should be: 8-frame models >  4-frame models > 1-frame models. Empirically, we observe a U-turn in the performance, and we believe the reason is the ground-truth data distribution of longer pseudo videos is more complex, and thus more expressive neural network architecture may be required to achieve optimal results (see Remark 4.2), while here we use the same UNet architecture across all models. Therefore, the results of 4-frame and 8-frame models may still be far from their optima. Ideally, if we are able to increase the flexibility of networks (e.g., depth and hidden dimension) as the length of pseudo videos grows, then according to our theory, the performance should always improve and after the model flexibility is enough to approximate the 1-frame data distribution well, further increasing the model flexibility will lead to saturating results for 1-frame but will keep improving the performance for 4-frame and 8-frame models. However, we keep the architecture the same across 1-frame, 4-frame and 8-frame models for a fair comparison in practice.
>
> >6. Related works.
>
> We will mention work on the data augmentation strategies in diffusion model contexts and the masked video-generation models in the related work section in our final revision. However, we would like to point out that our framework doesn’t rely on the masked video-generation framework and it can directly work with any image generative models that can be extended to video generative models.
>
> >7. Discussion about limitation.
>
> Indeed, video-generative models typically require more computing resources and increase the complexity compared with image-generative models. We will discuss these limitations in our final revision.

---

> ### Author Response · Authors · 2024-11-29
> **Response (part 2/2)**
>
> >8. Clarification in Theorem 4.1
>
> $s_1, s_2, …, s_k$ are time indices taken from $\{1, 2, …, T-1\}$, so $x_T$ is the last frame of the pseudo video, and $x_{s_1}, …, s_{s_k}$ are a subset of previous (corrupted) frames. $s_k > s_l$ indicates that $ \hat{x}_ {T}^{(g)} =g(x_{s_1}, x_{s_2}, ..., x_{s_k})$ predicts the last frame with previous frames taken from time step $s_1 > s_2 > … > s_k$ while   $\hat{x}_ T^{(h)}=h(x_{s_1}, x_{s_2}, \cdot\cdot\cdot, x_{s_l})$ predicts the last frame with previous frames taken from time step $s_1 > s_2 > … > s_k > … > s_l$. Therefore the first predictor uses fewer previous frames to predict the last frame. So in autoregressive generative models, to generate the last frame, we condition on previous frames at time steps $s_1 = T-1, s_2 = T-2, …, s_k = T-k, …, s_l = T-l$.
>
> >9. Clarification in Eq. 15.
>
> $x_T|x_{s_1},\cdot\cdot\cdot, x_{s_k} \,{\buildrel d \over =}\,  x_T|x_{s_1},\cdot\cdot\cdot, x_{s_l}$ means the conditional distributions $p(x_T|x_{s_1},\cdot\cdot\cdot, x_{s_k})$ and $p(x_T|x_{s_1},\cdot\cdot\cdot, x_{s_l})$ are the same.
>
> >10. Clarification for high-order Markov data augmentation on page 9.
>
> For first-order Markov, $x_ {T-t} = \sqrt{1-\beta_t} x_ {T-t+1} + \sqrt{\beta_t} \epsilon, \epsilon \sim \mathcal{N}(0, I)$, while for high-order Markov, $x_ {T-t} = \sqrt{1-\beta_t} [\frac{1}{t}\sum_{s=1}^{t}x_ {T-t+s}] + \sqrt{\beta_t} \epsilon$. We have added this expression in our uploaded revision.
>
> >11. How is the $F(h)$ matrix defined?
>
> $F(h)$ is the linear transformation corresponding to simulating the heat equation. And it has a similar effect as blurring the image. The reason why we use heat equation to blur the image is we can obtain a proper density for the forward process as in Eq. 11, while the standard blurring augmentation doesn’t give us closed-form density, which is required to train HVAE. We use T=18 here because we empirically find that with 18 steps of blurring using the heat equation, the MNIST images will visually be entirely blurred.
>
> > 12. What would happen if you only learned to tokenize the clean data (images) and then just tokenized the augmented frames when training the generative model?
>
> This is an interesting point. However, we believe pseudo-videos help train the tokenizer as well since we can see that the reconstruction of the clean images (the last frame) becomes strictly better as we increase the length of pseudo-videos (Table 1).
>
> >13. Code release.
>
> Yes, we plan to release our code upon publication of the paper. Currently, we are still tidying up the code.
>
> References
>
> [1] “Cold Diffusion: Inverting Arbitrary Image Transforms Without Noise”, NeurIPS 2023
>
> [2] “Phenaki: Variable Length Video Generation From Open Domain Textual Description”, ICLR 2023
>
> [3] “Transformers For Image Recognition At Scale”, ICLR 2021
>
> [4] “Escaping the Big Data Paradigm with Compact Transformers”, CVPR LLID Workshop 2021
>
> [5] “Transformers in Vision: A Survey”, ACM Computing Surveys 2022
>
> [6] “Discrete Predictor-corrector Diffusion Models for Image Synthesis”, ICLR 2023
>
> [7] “Language Model Beats Diffusion — Tokenizer Is Key To Visual Generation”, ICLR 2024

---

> > ### Comment · Reviewer_QFcD · 2024-12-06
> > **Thank you for your response**
> >
> > I thank the authors for their response. However, I'm not certain my concerns have been resolved, especially for the results with the artifacts. To my understanding, the authors use an architecture (pure ViTs) that does not work well for small datasets which results in bad reconstruction, compared to the training with videos which naturally provide more data. I find it hard to understand this design choice, as we have previous works that demonstrated excellent results for the 1-frame case. What is the point proven here? We have previous evidence for achitectures/models that work well for the 1-frame setting, but the authors choose to work with an architeture that does not work well, and build their entire case based upon this choice (and at least for that case, why not initialize the ViT with some-pretrained weights to get reasonable results for the 1-frame setting). The insight the authors provide in their response: "However, we believe pseudo-videos help train the tokenizer as well since we can see that the reconstruction of the clean images (the last frame) becomes strictly better as we increase the length of pseudo-videos (Table 1)" would be invalid, in my opinion, if the authors used a proper architecture and their 1-frame results matched the performance established in previous works. I just find that unconvincing though I like the idea of this paper.

---

> ### Author Response · Authors · 2024-12-06
> **Thank you for the follow-up**
>
> We thank the reviewer for the response.
> - The CViViT architecture is naturally a video-model extension of the ViT-based 1-frame VQVAE, and to our knowledge there is no video-model extension for pure CNN-based VQVAE in the literature, therefore, we consider using CViViT architecture when we design the experiments in Section 3.
> - The main purpose of the paper is not to achieve as high generation performance as possible. Rather, we are trying to verify our hypothesis on the possibility of improving the performance of image-generative models by extending them to their video-generative model counterparts. We understand that the 1-frame result in Section 3 may not match the CNN-based VQVAE, however, the experiments in Section 3 still provide evidence for our core claim: "It is possible to improve image generative models by jointly modelling the distribution of the original image and its corresponding pseudo video" (notice the motto of TMLR is "focusing on evidence to verify the paper claim rather than purely pursuing state-of-the-art"). By training on pseudo videos, we can improve an underperforming image-generative model significantly. The FID of the multi-frame models on CIFAR10 and CelebA are 35.5 and 16.6, respectively, and they match the performance of many well-tuned GAN models, which in general outperform the generation performance of standard VQVAE on these two datasets.
> - Furthermore, the observation that we may improve the generation performance of ViT-based models on small datasets by implicitly increasing the size of the data using pseudo videos itself can be considered as a beneficial by-product of pseudo video frameworks.
> - The comment about initializing the ViT with some pre-trained weights to get reasonable results for the 1-frame setting is very interesting. However, we are not aware that there is a pre-trained backbone of ViT-based generative model that can be directly applied over small images with 32 x 32 or 64 x 64 resolution. If the reviewer knows ViT-based pre-trained generative models that can be directly applied to CIFAR10 or CelebA, please let us know.

---

> > ### Comment · Reviewer_QFcD · 2024-12-06
> > **Thanks**
> >
> > I thank the authors for the continued engagement.
> > Please let me clarify my points.
> >
> > First, I like the idea of this paper and I am trying to get convinced that it indeed works as I believe it should work. However, I'm not convinced that the experiments and corresponding conclusions prove that. Mainly, I'm not asking the authors to run more experiments and I don't have any concerns regarding novelty, but merely to provide evidence for their claims which I find lacking (I'm familiar with TMLR's guidelines).
> >
> > The authors chose to use a ViT-based backbone and train it from scratch (below I refer to my comment regarding pre-trained backbones), knowing that they are going to test their hypothesis on small-scale data that is known to be hard to fit with pure ViTs.
> >
> > The authors test the hypothesis with the chosen architecture and find out that increasing the number of frames results in better generation performance, and from that they conclude that video models can be used for image generation. Now, I'm fully on-board with that and I'm not surprised that it works. However, **how can we tell that it is the video modeling itself and not just the additional data?** As the authors note, perhaps the reason for performance gains is merely data augmentation? I think it would not be a surprising result. Maybe with a proper architecture there would not be any gains (again, I want to believe that there would be). This is why I believe the sub-optimal choice of architecture may obscure the actual conclusion from this work. If the authors want to claim that additional data improves image generation with ViT as a backbone, then the results in the paper somewhat back that up. I think the fundamental design choices made in this paper are not supporting the hypothesis in question.
> >
> > As for my comment regarding pre-trained backbones -- I want to emphasize again that I'm not trying to get the authors to run more experiments, I was merely trying to think what I would do in your case (choosing to work with a ViT backbone). One of the first challenges the authors probably faced is the limited resources (which is completely understandable) which allows them to work with small-scale datasets (no complaints here!). I think the authors agree that it is known that pure ViTs are not great for that, and this is why the procedure, which has become standard in recent years, is utilizing a pre-trained model, use it to extract features, and then train (or even fine-tune) over these features. For example, it has become widely popular to use DINOv2 (https://github.com/facebookresearch/dinov2) to extract pre-trained features. As for the claim regarding the resolution, the simple resizing of the input image is quite common (and if you are not fine-tuning you can even pre-compute them to save memory). You can attach any architecture of choice over these pre-trained features, and I think this is something I would have at least tried just to get that "small-scale data" challenge out of the way. Again, not asking the authors to do that, just sharing my thoughts.
> >
> > Relying on previous work is important, but sometimes big architectures that are intended for big data are not optimal for academic purposes, and we need to make modifications to the design to fit our purposes (and the claim "this is what the used so we use it as well" is OK when the benchmark is the same, but this is not the case here, this paper tries to prove a point that does not require it to compare with the actual benchmark used with the original model).

---

> ### Author Response · Authors · 2024-12-07
>
> We thank the reviewer for the prompt response and detailed suggestions in the experimental setup, which are valuable for our future investigation in this line of research. Here we would like to address your concern about the CViViT experiments.
> - **Our core claim is “It is possible to improve image generative models by jointly modelling the distribution of the original image and its corresponding pseudo video which contains self-supervised information” (Sec 2.3)**, and our intuition is data augmentation in pseudo videos provides useful self-supervised information which helps the learning and generation. The key to the success of this framework is to design suitable data augmentation, which may vary for different types of generative models with different generative procedures (e.g., autoregressive or one-shot) and different architectures (e.g. Transformer or CNN-UNet). Although we theoretically analyse the benefits of High-order Markov data augmentation for autoregressive video generative models (Sec 4.1), in general, which data augmentations are optimal for different types of models are open research questions and we never claim we know the exact answer to it.
> -  The pseudo video framework is not just about having a video model. **The key insight in this framework is to improve generation performance by augmenting the image distribution to pseudo video distribution which contains useful self-supervised information of the original image.** In this sense, the performance improvement due to the additional data from creating pseudo videos in the experiments of Sec 3 is not against our claim since the implicitly increased data size in the pseudo video framework is a natural outcome of injecting more self-supervised information, and it has been empirically proven to be effective to improve the performance of CViViT models. Notice that useful data augmentation is also the key to the success of many self-supervised learning frameworks.
> - To demonstrate that **the data augmentation in the pseudo-video framework can provide more useful self-supervised information than standard 1-frame data augmentation techniques which also implicitly increase the data size**, we further train a 1-frame CViViT model on CIFAR10 but with a standard data augmentation technique in vision task: random horizontal flip, and the reconstruction FID and PSNR are 84.37 and 17.16, respectively, which are almost the same as our 1-frame result reported in the paper (FID: 84.25 & PSNR: 17.20). Other standard 1-frame data augmentation techniques such as random cropping and colour jittering are not considered here since they will change the training distribution (as opposed to pseudo video whose last frame is always from the true data distribution) and therefore will even hurt the generative performance (this is a limitation of standard 1-frame data augmentation techniques compared with our framework in generative model applications). We believe this suggests that in generative model applications, the pseudo video framework provides more useful self-supervised information than standard 1-frame data augmentation which also results in additional data.
>
> In summary, **similar to many self-supervised learning frameworks, the additional data from data augmentation is a natural outcome of injecting more self-supervised information and its effectiveness in improving generation performance aligns with our core claim**. Moreover, the pseudo video framework is not purely about having a video model (one can easily come up with bad design choices for data augmentation and even make the performance of video model worse than image model), its effectiveness depends heavily on whether the augmented distribution of pseudo videos contains useful self-supervised information that helps the learning and generation for the given model class. **We additionally provide evidence that the chosen data augmentation in Sec 3 in the pseudo-video framework is much more useful than standard 1-frame data augmentation which also increases the data size** (due to the time constraint of rebuttal, we only consider 1-frame data augmentation for CIFAR10 for now, but we will include the result for CelebA in our final revision as well).

---

### Review · Reviewer_k98F · 2024-11-14

**Summary Of Contributions:**

The authors study if generative image models can be improved by extending them to their corresponding generative _video_ models, and training these on "pseudo videos", i.e. sequences of images constructed by applying data augmentation to the original images (e.g., adding Gaussian noise or blur of increasing intensity to the original images).

They conduct experiments on the CIFAR10 (32x32 images) and CelebA (64x64 images) datasets, using multiple generative models.

They demonstrate that the image generation quality indeed can be improved by using this pseudo video-based approach, at least for some data augmentation methods and when using quite short pseudo videos.

**Audience:**

Yes

**Broader Impact Concerns:**

No concerns.

**Claims And Evidence:**

No

**Requested Changes:**

Overall, I quite like paper. I think the main idea is interesting and neat. I definitely think this could be relevant for the TMLR audience.

However, I think the current version requires some clarifications, see "Weaknesses" above.

**Strengths And Weaknesses:**

Strengths:
- The main idea of this paper, that generative image models could be improved by converting images into pseudo videos (inspired by the success of diffusion models) and training generative video models on these, is interesting and overall makes intuitive sense. It is a neat idea.
- The paper is well-written overall, most things are clearly described. Section 1 and 2 introduce the problem and proposed approach well.
- The experiments are quite encouraging overall, it does indeed seem to be possible to improve the image generation quality via the proposed pseudo video-based approach, at least for some settings, and to some extent.



Weaknesses:
- The two set of experiments in Section 3 and Section 4.2 are a bit confusing, they seem to give some conflicting conclusions.
- The aspect of computational cost could be discussed more explicitly.
- The results for "First-order Markov" in Table 2 seem a bit concerning, this sort of goes against the main idea of the paper. At the very least, I would not expect the pseudo video approach to significantly _downgrade_ the image generation quality for some settings.
- - Why do the "First-order Markov" downgrade the performance of the 1-frame model in Table 2, when it always significantly improves the results in Table 1? In Table 1, you even use Gaussian blur, which is shown to be worse than Gaussian noise in Table 4?



Other questions/suggestions:
- Only experiments on CIFAR10 and CelebA, could it be possible to at least get qualitative results on a dataset with more high-resolution images?
- In Table 1 and 2, are the 4/8/18-frame models 4/8/18x slower than the 1-frame model? In general, could you discuss the computational cost trade-off a bit more?
- In Section 3, why 8-frame and 18-frame models (the choice of 8 and 18 frames seems a bit random).
- Figure 2: _"Figure 2 demonstrates that the HVAE trained with pseudo videos created by the heat equation can generate much sharper and diverse digits than a standard HVAE"_, it is not really obvious to me that the images are 'much sharper'?
- Figure 3 is a neat example, makes you understand exactly what a "pseudo video" is. Could perhaps make sense to move this a bit earlier in the paper?
- For "Reconstruction" in Table 2, does the performance keep improving if you increase the number of frames further?
- In Section 3, could you perhaps compare the performance of Gaussian blur and Gaussian noise also in this setting? I think it could be interesting.



Minor things:
- Section 1: "may poses challenges" --> "may pose challenges"?
- Sec. 2.1: "by maximizing the the tractable", typo.
- "VQVAE" e.g. in Section 1 and Section 2.3, but then "VQ-VAE" in Section 3.

---

> ### Author Response · Authors · 2024-11-29
> **Response (part 1/3)**
>
> Thank you for your encouraging review and valuable feedback. We will address your questions below.
>
> >1. The two sets of experiments in Section 3 and Section 4.2 seem to give some conflicting conclusions.
>
> As we mentioned in Section 6, the design space of pseudo videos is huge in practice and it is likely that different model classes prefer pseudo videos based on different data augmentation strategies. Section 3 and Section 4.2 consider different types of video generative models. In section 3, we first compress the pseudo videos into latent discrete tokens with VQVAE and then a transformer-based model (VideoGPT or Phenaki) is used to fit a prior over latent tokens. In section 4, we consider a video diffusion model (FDM) that directly works on the data space instead of the latent space and it autoregressively generates the next frame in the pseudo video conditioned on a few previous frames. We will show that the difference in modelling choice suggests different types of pseudo videos are preferred in our response to your specific questions below.
>
> > 2. The aspect of computational cost could be discussed more explicitly.
>
> As we need to fit video models instead of image models, the computational cost increases and we will discuss this limitation in detail in the final revision.
>
> >3. Why does the "First-order Markov" downgrade the performance of the 1-frame model in Table 2, when it always significantly improves the results in Table 1?
>
> - The intuition of Theorem 4.1 fails for 1-st order Markov data augmentation (Table 2) in the following two ways:
>   * Theorem 4.1 suggests that the optimal performance of multi-frame autoregressive video-generative models trained on 1st-order Markov pseudo videos is the same no matter how long the videos are since 1st-order Markov satisfies the condition $p(x_t| x_{t-1}, x_{t-2}, …) = p(x_t|x_{t-1})$, which makes the equality hold. Therefore, the 4-frame models and the 8-frame models should have the same optimal performance if the network architecture is flexible enough. Intuitively, it means a larger length of the pseudo videos doesn’t necessarily help improve the performance.
>   * Theorem 4.1 is only concerned with the theoretically optimal performance, which is achievable only when the network is flexible enough (e.g., as universal function approximators). In practice, since we are using the same UNet architecture across 1-frame, 4-frame and 8-frame models and the data distribution is more complex with longer pseudo videos, it is likely that the practical performance can deviate from the theoretical optimum more for models trained on longer pseudo videos.
> - While in practice the gap from optimum still exists for high-order Markov due to the limitation of flexibility in the network architecture, with high-order Markov, the theoretically optimal performance will improve when the length of videos increases (the strict inequality in Theorem 4.1 holds in this case). Therefore, it is possible for us to achieve better performance with more frames in practice. However, with fixed architecture, the deviation from optimum due to the limitation in architecture tends to increase with longer pseudo videos whose data distribution is more complex. Therefore, in practice, we may observe a U-turn in performance with high-order Markov data augmentation if we keep the network architecture fixed.
> - Theorem 4.1 doesn’t apply to experiments in Section 3, since the CViViT is a Transformer-based architecture, it will mix the information of all the frames in the pseudo video into each latent token. Therefore, even with 1st-order Markov data augmentation, the procedure of first generating tokens in the latent space and then mapping it to the data space with a decoder is not an autoregressive video generation process in the data space anymore (which is assumed in Theorem 4.1). Therefore, even with first-order Markov data augmentation, it is possible the optimal performance with longer pseudo videos outperforms the optimal performance with shorter pseudo videos. Moreover, with data augmentation, we implicitly increase the size of the data and Transformers have been shown to be in favour of large data to work well in vision tasks [1,2,3], which can be another practical explanation of why the performance of multi-frame models outperform 1-frame models. However, we still see a U-turn for CIFAR10 in Table 1, which suggests for longer pseudo videos, more flexible architecture is needed to ensure the practical performance is close to optimum. In Table 1, we still keep the architecture to be the same across different models for a fair comparison.

---

> ### Author Response · Authors · 2024-11-29
> **Response (part 2/3)**
>
> >4.  In Table 1, you even use Gaussian blur, which is shown to be worse than Gaussian noise in Table 4?
>
> We did try using Gaussian noise as data augmentation initially for experiments in Section 3 and we found it was noticeably worse than using blurring in this case. We believe one of the reasons for this is we use a VQVAE to compress the data to discrete latent tokens here, and if using Gaussian noise as data augmentation, it will increase the variability of the pixel values in each patch. Therefore it would be hard to compress the pseudo videos with so much variability into discrete tokens defined with finite codebook size. In contrast, blurring will reduce the variability in the image and make the pixels close to each other have similar values, which makes it easier for a VQVAE with finite codebook size to compress the pseudo videos.
>
> In general, since the pseudo-video framework contains image-generative models as special cases (i.e., T=1), intuitively there will be a configuration of hyperparameters (type of data augmentation, architecture etc.) that make the performance better than image (1-frame) generation. However, as we mentioned in Section 6, pseudo-video frameworks make the space of hyperparameters larger, especially the design space of data augmentation for creating pseudo-videos is huge. For different types of models and architectures, the optimal data augmentation is in general not the same. While we have theoretically shown some intuition that high-order Markov data augmentation is preferable in autoregressive video generation models, finding optimal data augmentation or practical suggestions of what data augmentation is more suitable for other types of models remains an open research question. Nevertheless, we believe our analysis and empirical experiments have demonstrated a proof of concept that this is a promising framework with a lot of potential to improve generation performance.
>
> >5. Results on a dataset with more high-resolution images.
>
> Unfortunately, the scale of experiments presented in the paper is the largest one we can consider with our computing resources. We would be interested in doing larger-scale experiments in the future if more computing resources are available to us.
>
> >6. In Table 1 and 2, are the 4/8/18-frame models 4/8/18x slower than the 1-frame model? In general, could you discuss the computational cost trade-off a bit more?
>
> For VQVAE-based experiments in Section 3 (Table 1), the 8/18-frame models are roughly 4/9 x slower than the 1-frame model since for multi-frame models, CViViT will compress two consecutive frames into one in the latent space: as we mention in “Network Architectures” in page 6, a pseudo video with shape (T, H, W, C), will be compressed to tokens with shape (T/2, H/4, W/4, C) and an image with shape (1, H, W, C) will be compressed to tokens with shape (1, H/4, W/4, C). For experiments based on video diffusion in Section 4.2 (Table 2), the 4/8-frame models are roughly 3/5x slower than the 1-frame model since they use overall 3/5k diffusion steps, while the 1-frame model uses 1k diffusion steps overall. For 1-frame models, we also try using 4k diffusion steps (Table 3) so that they have a bit worse sampling efficiency than the 4-frame models. While it improves the performance of the 1-frame model for CIFAR10, it doesn’t improve performance for CelebA.
>
> >7. In Section 3, why 8-frame and 18-frame models?
>
> During our experiments, we first choose the largest number of frames to be the largest one that we can train on our GPU (they are 18 and 8 for CViViT and Video Diffusion, respectively). Then we consider including the results with the number of frames approximately being half of the maximal one (8 and 4 for CViViT and Video Diffusion, respectively).
>
> >8. Comparison in Figure 2.
>
> We will rephrase accordingly. We further report the Inception score (computed based on a CNN MNIST classifier) as a quantitative metric for the generation and heat HVAE outperforms standard HVAE. We have added this result in our uploaded revision.
>
> ||Standard HVAE|Heat HVAE|
> |---|---|---|
> |Inception Score (IS)| 7.37| **9.32**|
>
> >9. Move Figure 3 a bit earlier in the paper?
>
> Thank you for the suggestion, we will rearrange its location in our final revision.
>
> > 10. For "Reconstruction" in Table 2, does the performance keep improving if you increase the number of frames further?
>
> As we mentioned in our response to Q7, 18 is the maximal number of frames we can fit into our GPU, therefore, we haven’t tried using more frames. However, since we already see a diminishing return in Table 1, we think it is likely the reconstruction performance will saturate or we may even see a U-turn if we keep increasing the number of frames.

---

> ### Author Response · Authors · 2024-11-29
> **Response (part 3/3)**
>
> >11.  In Section 3, could you perhaps compare the performance of Gaussian blur and Gaussian noise also in this setting?
>
> As we mentioned in our response to Q4, We did try using Gaussian noise as data augmentation initially for experiments in Section 3 and we found it was noticeably worse than using blurring in this case.
>
> >12. Typos
>
> We thank the reviewer for pointing out these typos, we have revised them in our uploaded revision.
>
> References
>
> [1] “Transformers For Image Recognition At Scale”, ICLR 2021
>
> [2] “Escaping the Big Data Paradigm with Compact Transformers”, CVPR LLID Workshop 2021
>
> [3] “Transformers in Vision: A Survey”, ACM Computing Surveys 2022

---

> > ### Comment · Reviewer_k98F · 2024-12-04
> >
> > I have read the other reviews and all author responses.
> >
> > The other reviews are overall quite similar to mine (the idea is neat/interesting but the experiments could be more convincing/clear).
> >
> > The author responses address some concrete questions, but I still find the experimental results somewhat unclear / not overly convincing.
> >
> > I'm a bit borderline on this paper. But I do like the main idea and I definitely think it could be relevant for the TMLR audience.
> >
> > As a proof of concept, this is probably good enough. I'm leaning towards accept.

---

> > > ### Comment · Reviewer_k98F · 2024-12-08
> > >
> > > I have read the continued discussion between the authors and reviewer QFcD.
> > >
> > > I think the authors respond quite well overall.
> > >
> > > This is definitely not a perfect paper, but I like the main idea and it's definitely relevant for the TMLR audience. As a proof of concept paper, I think this is good enough.
> > >
> > > I will select "Leaning Accept" as my Decision Recommendation.

---

### Review · Reviewer_v4gG · 2024-11-21

**Summary Of Contributions:**

The paper suggests that using pseudovideos, which consist of frames generated during the forward process of diffusion models, can enhance the performance of certain generative models. Furthermore, the authors empirically demonstrate that employing higher-order Markov pseudovideos improves the performance of Improved DDPM.

**Audience:**

Yes

**Claims And Evidence:**

No

**Requested Changes:**

See weaknesses.

**Strengths And Weaknesses:**

pseudovideos improves the performance of Improved DDPM.

Strengths: The authors present an interesting perspective on why diffusion models achieve remarkable performance, analyzed through the framework of other generative models.

Weaknesses: Although the authors have presented a number of experiments, I believe that additional experiments would further strengthen the paper and provide a more comprehensive evaluation.

1. Contributions: While the contributions outlined in the introduction are interesting, I feel that they could be presented more clearly in the paper. My understanding is that Section 2 should provide a stronger motivation for why pseudovideos would improve the performance of HVAE. This seems to be primarily addressed by the MNIST experiment, which compares two HVAE models, one with a fixed encoder according to the heat equation. However, the comparison seems to rely solely on visual results, without including metrics that would better support and convey the argument. The inclusion of such metrics could significantly strengthen the presentation.

2. Experiments:  I believe that additional experiments would further strengthen the paper and provide a more comprehensive evaluation.


- In particular, I believe that the inclusion of additional datasets in this empirical investigation could provide a more comprehensive analysis. How well would this approach would scale with higher resolution images?  I also wonder if this approach might be more computationally expensive to train. Could you elaborate on the trade-offs involved?

- Section 3: Could the authors clarify why there is no clear evidence as to whether 18-frame or 8-frame pseudovideos are more appropriate? Are these differences due to the specific characteristics of the networks? In addition, could the authors provide the FID scores of the trained DDPM models on the same datasets to provide a more complete comparison?

- Section 4: From my review, the FID score of the Improved DDPM reported in this paper does not match the results of the original paper. It might be valuable to replicate the original results and evaluate how this would affect the pseudovideo counterparts.
- Section 4: My understanding is that the applied higher order Markov noise schedule reduces the noise in the pseudovideo frames due to the averaging process, which likely explains the observed performance improvement. However, it is surprising that the best performance is not achieved with 8 frames. The authors suggest that this may be related to the expressiveness of the networks. Have the authors explored different network architectures to investigate potential performance improvements? How does the sampling efficiency of the new approach compare to that of the Improved DDPM? In addition, how might a VQVAE-based network perform when applied to this particular type of pseudovideo?

---

> ### Author Response · Authors · 2024-11-29
> **Response (part 1/2)**
>
> Thank you for your review and valuable feedback. We will address your questions below.
>
> >1. The comparison in Section 2 seems to rely solely on visual results, without including metrics that would better support and convey the argument.
>
> We further report the Inception score (computed based on a CNN MNIST classifier) as a quantitative metric for the generation and heat HVAE outperforms standard HVAE. We have added this result in our uploaded revision.
>
> ||Standard HVAE|Heat HVAE|
> |---|---|---|
> |Inception Score|7.27|**9.32**|
>
> >2.  Inclusion of additional datasets in this empirical investigation.
>
> Unfortunately, the scale of experiments presented in the paper is the largest one we can consider with our computing resources. We would be interested in doing larger-scale experiments in the future if more computing resources are available to us.
>
> >3. Section 3: Could the authors clarify why there is no clear evidence as to whether 18-frame or 8-frame pseudo-videos are more appropriate?
>
> During our experiments, we first choose the largest number of frames to be the largest one that we can train on our GPU (they are 18 and 8 for CViViT and Video Diffusion, respectively). Then we consider including the results with the number of frames approximately being half of the maximal one (8 and 4 for CViViT and Video Diffusion, respectively).
>
> >4. Could the authors provide the FID scores of the trained DDPM models on the same datasets to provide a more complete comparison?
>
> We would like to clarify that the experiments in Section 4 are based on video diffusion directly working in the data space instead of the latent space as for the VQVAE framework presented in Section 3. Therefore, it requires more computing resources and the maximal length (8) of the pseudo videos we are able to consider in Section 4 is shorter given our computing resources. Although for different model classes, the suitable data augmentation may vary due to computation budget or characteristics of network architectures, the interesting observation here is for two different classes of models, in practice, we are able to find some simple data augmentation strategies to improve the performance of generation within the pseudo-video frameworks.
>
> >5. The FID score of the Improved DDPM reported in this paper does not match the results of the original paper.
>
> The experiments are based on the official Implementation of Improved DDPM. However, due to our computing budget, we have to consider smaller-scale experiments than the original paper. The original paper considers a larger UNet architecture with 3 residual blocks and here we consider a smaller UNet with 2 residual blocks so that we can fit the 8-frame model into our GPU for CelebA. Moreover, they consider 4k diffusion steps while we only consider 1k diffusion steps. The FID reported in the original paper is computed based on 50k samples while ours is computed based on 10k samples.
>
> >6. (Section 4). The applied higher-order Markov noise schedule reduces the noise in the pseudo-video frames due to the averaging process, which likely explains the observed performance improvement.
>
> During the hyperparameter selection stage, we experiment with 1st-order Markov Gaussian noise data augmentation with the maximal $\beta$ (controlling noise level) selected from {0.02, 0.05, 0.1}, and the 0.05 is the one that leads to the best performance with 1st-order Markov Gaussian noise augmentation. We keep the maximal $\beta$ to be 0.05 for high-order Markov Gaussian noise augmentation without further tuning it. Therefore, the model trained on less-noisy 1srt-order Markov pseudo videos (max $\beta$ = 0.02) is worse than what we report in the paper and we believe this suggests the noise reduction due to the averaging process is not a key factor to explain the improved performance for high-order Markov data augmentation.

---

> ### Author Response · Authors · 2024-11-29
> **Response (part 2/2)**
>
> >7. (Section 4). It is surprising that the best performance is not achieved with 8 frames.
>
> If the model class is fully expressive such that we can reach the theoretical optima in Theorem 4.1 for 1-frame, 4-frame and 8-frame, then the rank of the theoretically optimal performance should be: 8-frame models >  4-frame models > 1-frame models. Empirically, we observe a U-turn in the performance, and we believe the reason is the ground-truth data distribution of longer pseudo videos is more complex, and thus more expressive neural network architecture may be required to achieve optimal results (see Remark 4.2), while here we use the same UNet architecture across all models. Therefore, the results of 4-frame and 8-frame models may still be far from their optima. Ideally, if we are able to increase the flexibility of networks (e.g., depth and hidden dimension) as the length of pseudo videos grows, then according to our theory, the performance should always improve and after the model flexibility is enough to approximate the 1-frame data distribution well, further increasing the model flexibility will lead to saturating results for 1-frame but will keep improving the performance for 4-frame and 8-frame models. However, we keep the architecture the same across 1-frame, 4-frame and 8-frame models for a fair comparison in practice. The current architecture is the largest we can consider to fit the 8-frame model with our computing resources.
>
> >8. How does the sampling efficiency of the new approach compare to that of the Improved DDPM?
>
> The 4/8-frame models are roughly 3/5x slower than the 1-frame model since they use overall 3/5k diffusion steps, while the 1-frame model uses 1k diffusion steps overall. For 1-frame models, we also try using 4k diffusion steps (Table 3) so that they have a bit worse sampling efficiency than the 4-frame models. While it improves the performance of the 1-frame model for CIFAR10, it doesn’t improve performance for CelebA.
>
> >9. How might a VQVAE-based network perform when applied to this particular type of pseudo video?
>
> We did try using Gaussian noise as data augmentation initially for experiments in Section 3 (based on VQVAE) and we found it was noticeably worse than using blurring in this case. We believe one of the reasons for this is that VQVAE compresses the data to discrete latent tokens, and if using Gaussian noise as data augmentation, it will increase the variability of the pixel values in each patch. Therefore it would be hard to compress the pseudo videos with so much variability into discrete tokens defined with finite codebook size. In contrast, blurring will reduce the variability in the image and make the pixels close to each other have similar values, which makes it easier for a VQVAE with finite codebook size to compress the pseudo videos.
>
> In general, since the pseudo-video framework contains image-generative models as special cases (i.e., T=1), intuitively there will be a configuration of hyperparameters (type of data augmentation, architecture etc.) that make the performance better than image (1-frame) generation. However, as we mentioned in Section 6, pseudo-video frameworks make the space of hyperparameters larger, especially the design space of data augmentation for creating pseudo-videos is huge. For different types of models and architectures, the optimal data augmentation is in general not the same. While we have theoretically shown some intuition that high-order Markov data augmentation is preferable in autoregressive video generation models, finding optimal data augmentation or practical suggestions of what data augmentation is more suitable for other types of models remains an open research question. Nevertheless, we believe our analysis and empirical experiments have demonstrated a proof of concept that this is a promising framework with a lot of potential to improve generation performance.

---

> > ### Comment · Reviewer_v4gG · 2024-12-16
> >
> > Thanks to the authors for their detailed responses and engaging discussions. Like the other reviewers, I find the proposed idea interesting. However, I am still not convinced by the experimental results and have concerns about the computational cost of this method compared to 1-frame samplers. I do not think that these concerns have been adequately addressed in the authors' responses. Therefore, my score remains borderline.

---

### Author Response · Authors · 2024-11-29
**General Response**

We thank all reviewers for their valuable feedback. We have responded to each reviewer's comments separately below their respective review and updated our manuscript (with changes highlighted in blue). Specifically, in the updated revision, we include:
- Inception Score (IS) as a quantitative metric to evaluate the heat HVAE vs standard HVAE trained on MNIST in Section 2 (Table 1 in Section 2),
- Peak Signal-to-noise Ratio (PSNR) as an additional metric to evaluate the reconstruction of CViViTs in Section 3 (Table 6 in Appendix C),
- the explicit expressions for first-order Markov & High-order Markov data augmentation strategies discussed in Section 4.2.
- details about hyperparameters in Appendix D.

We hope that this along with our responses to each reviewer has sufficiently addressed their concerns.

---

### Decision · Action_Editor_GXRs · 2025-01-09

**Recommendation:** Reject

**Comment:**

I face a very hard decision to make because the paper is very interesting is definitely worth accepting. Moreover, TMLR has a clear policy about experiments (i.e., no SOTA chasing). Nevertheless, it is crucial to support the claims with appropriate experimental validation. It seems this is missing in the current version of the paper. Moreover, all three reviewers raised doubts about the experimental section and they are torn (2 leaning rejection vs. 1 leaning acceptance). I am afraid I have to agree with the 2 leaning rejection reviewers. The paper would gain a lot if experiments are updated and designed as suggested by the reviewers.

Therefore, I recommend a reject but the authors are encouraged to submit a major revision at a later time.

**Audience:**

All reviewers agree that the paper is relevant to TMLR's  audience. There is no doubt about it.

**Claims And Evidence:**

The paper poses the following claims:
- self-supervised learning using pseudovideos can enhance the performance of certain generative models;
- employing higher-order Markov pseudovideos improves the performance of diffusion-based models.

Overall, the reviewers agree that the paper is interesting and the provided theory is sound. However, there are many concerns regarding the empirical validation of the idea. All reviewers raised the following points:
- what about the computational cost of this method compared to 1-frame samplers;
- the experimental results are however somewhat unclear / not overly convincing;
- the claims are not proven with the experimental benchmark used in the paper, namely, the architecture used in the paper (pure vision transformer) is known to under-perform in limited data settings (such as CIFAR10 and CelebA) when trained from scratch, thus, adding more data should help; in that sense, it is unclear whether the video modeling itself (i.e. the temporal modeling) or just the additional amount of data is responsible for the improvement of the performance.

**Resubmission Of Major Revision:**

The authors may consider submitting a major revision at a later time.